# Smart Multi-tenant Federated Learning

## Abstract

Federated learning (FL) is an emerging distributed machine learning method that empowers in-situ model training on decentralized edge devices. However, multiple simultaneous training activities could overload resource-constrained devices. In this work, we propose a smart multi-tenant FL system, MuFL, to effectively coordinate and execute simultaneous training activities. We first formalize the problem of multi-tenant FL, define multi-tenant FL scenarios, and introduce a vanilla multi-tenant FL system that trains activities sequentially to form baselines. Then, we propose two approaches to optimize multi-tenant FL: 1) *activity consolidation* merges training activities into one activity with a multi-task architecture; 2) after training it for rounds, *activity splitting* divides it into groups by employing affinities among activities such that activities within a group have better synergy. Extensive experiments demonstrate that MuFL outperforms other methods while consuming 40% less energy. We hope this work will inspire the community to further study and optimize multi-tenant FL.

## 1 Introduction

Federated learning (FL) (McMahan et al., 2017) has attracted considerable attention as it enables privacy-preserving distributed model training among decentralized devices. It is empowering growing numbers of applications in both academia and industry, such as Google Keyboard (Hard et al., 2018), medical imaging analysis (Li et al., 2019; Sheller et al., 2018), and autonomous vehicles (Zhang et al., 2021a; Posner et al., 2021). Among them, some applications contain multiple training activities for different tasks. For example, Google Keyboard includes query suggestion (Yang et al., 2018), emoji prediction (Ramaswamy et al., 2019), and next-world prediction (Hard et al., 2018); autonomous vehicles relates to multiple computer vision (CV) tasks, including lane detection, object detection, and semantic segmentation (Janai et al., 2020).

However, multiple simultaneous training activities could overload edge devices (Bonawitz et al., 2019). Edge devices have tight resource constraints, whereas training deep neural networks for the aforementioned applications is resource-intensive. As a result, the majority of edge devices can only support one training activity at a time (Liu et al., 2019); multiple simultaneous federated learning activities on the same device could overwhelm its memory, computation, and power capacities. Thus, it is important to navigate solutions to well coordinate these training activities.

A plethora of research on FL considers only one training activity in an application. Many studies are devoted to addressing challenges including statistical heterogeneity (Li et al., 2020; Wang et al., 2020a), system heterogeneity (Chai et al., 2020; Yang et al., 2021), communication efficiency (Karimireddy et al., 2020; Zhu et al., 2021), and privacy issues (Bagdasaryan et al., 2020; Huang et al., 2021). A common limitation is that they only focus on one training activity, but applications like Google Keyboard and autonomous vehicles require multiple training activities for different targets (Yang et al., 2018; Ramaswamy et al., 2019). Multi-tenancy of an FL system is designed by Bonawitz et al. (2019) to prevent simultaneous training activities from overloading devices. However, it mainly considers differences among training activities, neglecting potential synergies.

In this work, we propose a smart multi-tenant federated learning system, MuFL, to efficiently coordinate and execute simultaneous training activities under resource constraints by considering both synergies and differences among training activities. We first formalize the problem of multi-tenant FL and define four multi-tenant FL scenarios based on two variances in Section 3: 1) whether all training activities are the same type of application, e.g., CV applications; 2) whether all clients

support all training activities. Then, we define a vanilla multi-tenant FL system that supports all scenarios by training activities sequentially. Built on it, we further optimize the scenario, where all training activities are the same type and all clients support all activities, by considering both synergies and differences among activities in Section 4. Specifically, we propose *activity consolidation* to merge training activities into one activity with a multi-task architecture that shares common layers and has specialized layers for each activity. We then introduce *activity splitting* to divide the activity into multiple activities based on their synergies and differences measured by affinities between activities.

We demonstrate that MuFL reduces the energy consumption by over 40% while achieving superior performance to other methods via extensive experiments on three different sets of training activities in Section 5. We believe that MuFL is beneficial for many real-world applications such as autonomous vehicles, voice assistance systems, and robotics (more examples in Appendix A). We summarize our contributions as follows:

- We formalize the problem of multi-tenant FL and define four multi-tenant FL scenarios. To the best of our knowledge, we are the first work that investigates multi-tenant FL in-depth.
- We propose MuFL, a smart multi-tenant federated learning system to efficiently coordinate and execute simultaneous training activities by proposing activity consolidation and activity splitting to consider both synergies and differences among training activities.
- We establish baselines for multi-tenant FL and demonstrate that MuFL elevates performance with significantly less energy consumption via extensive empirical studies.

## 2 RELATED WORK

In this section, we first review the concept of multi-tenancy in cloud computing and machine learning. Then, we provide a literature review of multi-task learning and federated learning.

**Multi-tenancy of Cloud Computing and Machine Learning**  Multi-tenancy has been an important concept in cloud computing. It refers to the software architecture where a single instance of software serves multiple users (Chong & Carraro, 2006; Fehling et al., 2010). Multi-tenant software architecture is one of the foundations of software as a service (SaaS) applications (Mietzner et al., 2008; Cai et al., 2013). Recently, researchers have adopted this idea to machine learning (especially deep learning) training and inference. Specifically, some studies investigate how to share GPU clusters among multiple users to train deep neural networks (DNN) (Jeon et al., 2019; Zhao et al., 2020; Lao et al., 2021), but these methods are for GPU clusters that have enormous computing resources, which are inapplicable to edge devices that have limited resources. Targeting on-device deep learning, some researchers define multi-tenant as processing multiple computer vision (CV) applications for multiple *concurrent* tasks (Fang et al., 2018; Jiang et al., 2018). However, they focus on the multi-tenant on-device *inference* rather than training. On the contrary, we focus on multi-tenant federated learning (FL) *training* on devices, where the multi-tenancy refers to multiple concurrent FL training activities.

**Multi-task Learning**  Multi-task learning is a popular machine learning approach to learn models that generalize on multiple tasks (Thrun, 1995; Zhang & Yang, 2021). A plethora of studies investigate parameter sharing approaches that share common layers of a similar architecture (Caruana, 1997; Eigen & Fergus, 2015; Bilen & Vedaldi, 2016; Nekrasov et al., 2019). Besides, many studies employ new techniques to address the negative transfer problem (Kang et al., 2011; Zhao et al., 2018) among tasks, including soft parameter sharing (Duong et al., 2015; Misra et al., 2016), neural architecture search (Lu et al., 2017; Huang et al., 2018; Vandenhende et al., 2019; Guo et al., 2020; Sun et al., 2020), and dynamic loss reweighting strategies (Kendall et al., 2018; Chen et al., 2018; Yu et al., 2020). Instead of training all tasks together, task grouping trains only similar tasks together. The early works of task grouping (Kang et al., 2011; Kumar & Daumé, 2012) are not adaptable to DNN. Recently, several studies analyze the task similarity (Standley et al., 2020) and task affinities (Fifty et al., 2021) for task grouping. In this work, we adopt the idea of task grouping to consolidate and split training activities. The state-of-the-art task grouping methods (Standley et al., 2020; Fifty et al., 2021), however, are unsuitable for our scenario because they focus on the inference efficiency, bypassing the intensive computation on training. Thus, we propose activity consolidation and activity splitting to group training activities based on their synergies and differences.

**Federated Learning**  Federated learning emerges as a promising privacy-preserving distributed machine learning technique that uses a central server to coordinate multiple decentralized clients to

train models (McMahan et al., 2017; Kairouz et al., 2021). The majority of studies aim to address the challenges of FL, including statistical heterogeneity (Li et al., 2020; Wang et al., 2020a;b; Zhuang et al., 2020; yuyang deng et al., 2021; Zhang et al., 2021b), system heterogeneity (Chai et al., 2020; Yang et al., 2021), communication efficiency (McMahan et al., 2017; Konečný et al., 2016; Karimireddy et al., 2020; Zhu et al., 2021), and privacy concerns (Bagdasaryan et al., 2020; Huang et al., 2021). Among them, federated multi-task learning (Smith et al., 2017; Marfoq et al., 2021) is an emerging method to learn personalized models to tackle statistical heterogeneity. However, these personalized FL methods mainly focus on training one activity of an application in a client. Multi-tenant FL that handles multiple concurrent training activities is rarely discussed; the prior work (Bonawitz et al., 2019) mainly considers the differences among training activities. In this work, we optimize multi-tenant FL by further considering their synergies by splitting activities into groups. Our problem is also fundamentally different from clustered FL (Ghosh et al., 2020; Ouyang et al., 2021); They group models of the same training activity, whereas we group training activities.

## 3 PROBLEM SETUP

This section provides preliminaries of federated learning (FL), presents the problem definition of multi-tenant FL, and classifies four multi-tenant FL scenarios. Besides, we introduce a vanilla multi-tenant FL system supports for all scenarios.

### 3.1 PRELIMINARIES AND PROBLEM DEFINITION

In the federated learning setting, the majority of studies consider optimizing the following problem:

$$\min_{\omega \in \mathbb{R}^d} f(\omega) := \sum_{k=1}^{K} p_k f_k(\omega) := \sum_{k=1}^{K} p_k \mathbb{E}_{\xi_k \sim \mathcal{D}_k}[f_k(\omega; \xi_k)], \tag{1}$$

where $\omega$ is the optimization variable, $K$ is the number of selected clients to execute training, $f_k(\omega)$ is the loss function of client $k$, $p_k$ is the weight of client $k$ in model aggregation, and $\xi_k$ is the training data sampled from data distribution $\mathcal{D}_k$ of client $k$. FedAvg (McMahan et al., 2017) is a popular federated learning algorithm, which sets $p_k$ to be proportional to the dataset size of client $k$.

Equation 1 illustrates the objective of single training activity in FL, but in real-world scenarios, multiple simultaneous training activities could overload edge devices. We further formalize the problem of multi-tenant FL as follows.

In multi-tenant FL, a server coordinates a set of clients $\mathcal{C}$ to execute a set of $n$ FL training activities $\mathcal{A} = \{\alpha_1, \alpha_2, \ldots, \alpha_n\}$. It obtain a set of parameters of models $\mathcal{W} = \{\omega_1, \omega_2, \ldots, \omega_n\}$, where each model $\omega_i$ is for activity $\alpha_i$. By defining $\mathcal{M}(\alpha_i; \omega_i)$ as performance measurement of each training activity $\alpha_i$, multi-tenant FL aims to maximize the performance of all training activities $\sum_{i=1}^{n} \mathcal{M}(\alpha_i; \omega_i)$, under the constraint that each client $k$ has limited memory budget and computation budget. These budgets constrain the number of concurrent training actvitities $n_k$ on client $k$. Besides, as devices have limited battery life, we would like to minimize the energy consumption and training time to obtain $\mathcal{W}$ from training activities $\mathcal{A}$.

### 3.2 MULTI-TENANT FL SCENARIOS

We classify multi-tenant FL into four different scenarios based on variances in two aspects: 1) whether all training activities in $\mathcal{A}$ are the same type of application, e.g., computer vision (CV) applications or natural language processing (NLP) applications; 2) whether all clients in $\mathcal{C}$ support all training activities in $\mathcal{A}$. We depict these four scenarios in Figure 6 in Appendix A and describe them below.

**Scenario 1** $\forall \alpha_i \in \mathcal{A}$, $\alpha_i$ is the same type of application; $\forall \alpha_i \in \mathcal{A}$, $\forall c_k \in \mathcal{C}$ supports $\alpha_i$. For example, autonomous vehicles (clients) support the same sets of CV applications, such as object detection and semantic segmentation. Thus, they support training activities of these applications.

**Scenario 2** $\exists \alpha_i \in \mathcal{A}$, $\alpha_i$ is a different type of application; $\forall \alpha_i \in \mathcal{A}$, $\forall c_k \in \mathcal{C}$ supports $\alpha_i$. For example, Google Keyboard has different types of applications, including recommendation (query suggestion (Yang et al., 2018)) and NLP (next-world prediction (Hard et al., 2018)). Mobile phones (clients) with Google Keyboard support these applications together with all related training activities.

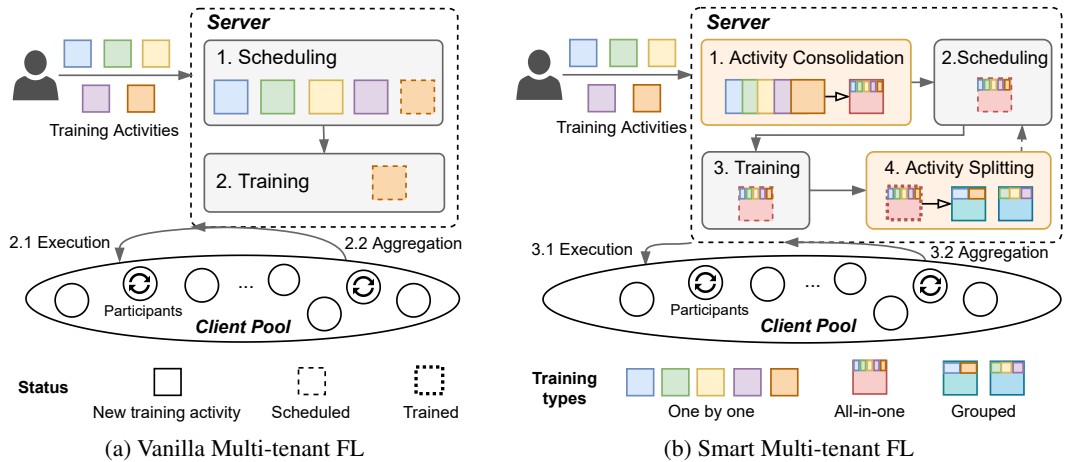

Figure 1: The architectures of proposed multi-tenant federated learning (FL) systems. The vanilla multi-tenant FL system (a) employs a scheduler to queue simultaneous training activities and execute them *one by one*. The smart multi-tenant FL system (b) proposes activity consolidation and activity splitting to consider both synergies and differences among training activities, which elevates performance and reduces resource consumption.

**Scenario 3** $\forall \alpha_i \in \mathcal{A}$, $\alpha_i$ is the same type of application; $\forall \alpha_i \in \mathcal{A}$, $\exists c_k \in \mathcal{C}$ does not support $\alpha_i$. For example, survelliance cameras (clients) in parking lots could support CV applications, but cameras in different locations may support different applications, e.g., counting open-parking spots, tracking parking duration, or recording fender benders.

**Scenario 4** $\exists \alpha_i \in \mathcal{A}$, $\alpha_i$ is a different type of application; $\forall \alpha_i \in \mathcal{A}$, $\exists c_k \in \mathcal{C}$ does not support $\alpha_i$. For example, browsers (clients) could leverage users' browsing history to support ranking (Hartmann et al., 2019) and news recommendation (Minto et al., 2021), which are different applications. Users may opt-out of recommendations, resulting in not all browsers supporting all training activities.

The application determines the multi-tenant FL scenario. We next introduce a vanilla multi-tenant FL that supports all these scenarios as our baseline.

### 3.3 VANILLA MULTI-TENANT FL

Figure 1a presents the architecture of a vanilla multi-tenant FL system. It prevents overloading and congestion of multiple simultaneous training activities by scheduling them to execute *one by one*. Particularly, we use a scheduler to queue training activities in the server (e.g., First in, First out). In each round, the server selects $K$ clients from the client pool to participate in training. The number of simultaneous training activities depend on the computational resources of the selected clients. In this study, we assume that each client can execute one training activity at a time ($n_k = 1$). This is a realistic assumption for the majority of current edge devices. [1] As a result, the vanilla multi-tenant FL system executes training activities sequentially.

The vanilla multi-tenant FL system supports the four multi-tenant FL scenarios. From the perspective of the type of application, it can handle different application types of training activities as each training activity is executed independently. From the perspective of whether clients support all training activities, each training activity can select clients that support the activity to participate in training. Despite its comprehensiveness, it only considers differences among training activities, neglecting their potential synergies. In contrast, our proposed MuFL considers both synergies and differences among training activities to further optimize the Scenario 1.

---

[1] Edges devices, e.g., NVIDIA Jetson TX2 and AGX Xavier, have only one GPU; GPU virtualization (Hong et al., 2017) that enables concurrent training on the same GPU currently are mainly for the cloud stack.

## 4   SMART MULTI-TENANT FL

In this section, we introduce the smart multi-tenant FL system, MuFL. We start by providing an overview of MuFL. Then, we present two important components of MuFL, activity consolidation and activity splitting, to consider both synergies and differences among simultaneous training activities.

Figure 1b depicts the architecture and training processes of MuFL. It contains a server to coordinate training activities and a pool of clients to execute training. MuFL optimizes the Scenario 1 discussed previously with the following steps: 1) The server receives training activities $\mathcal{A} = \{\alpha_1, \alpha_2, \ldots, \alpha_n\}$ to train models $\mathcal{W} = \{\omega_1, \omega_2, \ldots, \omega_n\}$ and *consolidates* these activities into an all-in-one training activity $\alpha_0$; 2) The server schedules $\alpha_0$ to train; 3) The server select $K$ clients from the client pool to execute $\alpha_0$ iteratively through FL process for $R_0$ rounds; 4) The server *splits* the all-in-one activity $\alpha_0$ into multiple training activity groups $\{\mathcal{A}_1, \mathcal{A}_2, \ldots\}$, where each group trains nonoverlapping subset of $\mathcal{W}$; the number of groups can be determined by the inference budget for the number of concurrent models. 5) The server iterates step 2 and 3 to train $\mathcal{A}_j$. We summarize MuFL in Algorithm 1 in Appendix D and introduce the details of activity consolidation and activity splitting next.

### 4.1   ACTIVITY CONSOLIDATION

Focusing on optimizing the Scenario 1 of multi-tenant FL, we first propose activity consolidation to consolidate multiple training activities into an all-in-one training activity, as illustrated in the first step of Figure 1b. In Scenario 1, all training activities are the same type of application and all clients support all training activities. Since training activities $\mathcal{A} = \{\alpha_1, \alpha_2, \ldots, \alpha_n\}$ are of the same type, e.g., CV or NLP, models $\mathcal{W} = \{\omega_1, \omega_2, \ldots, \omega_n\}$ could share the same backbone (they share the same encoder but could have different decoders). Thus, we can consolidate $\mathcal{A}$ into an all-in-one training activity $\alpha_0$ that trains a multi-task model $\nu = \{\theta_s\} \cup \{\theta_{\alpha_i} | \alpha_i \in \mathcal{A}\}$, where $\theta_s$ is the shared model parameters and $\theta_{\alpha_i}$ is the specific parameters for training activity $\alpha_i \in \mathcal{A}$. The loss function for all-in-one training is $\mathcal{L}(\mathcal{X}, \theta_s, \{\theta_{\alpha_i}\}) = \sum_{\alpha_i \in \mathcal{A}} \mathcal{L}_{\alpha_i}(\mathcal{X}, \theta_s, \{\theta_{\alpha_i}\})$.

Activity consolidation leverages synergies among training activities and effectively reduces the computation cost of multi-tenant FL from multiple trainings into a single training. However, simply employing activity consolidation is another extreme of multi-tenant FL that only considers synergies among activities. As shown in Figure 2, all-in-one method is efficient in energy consumption, but it leads to unsatisfactory performance. Consequently, we further propose activity splitting to consider both synergies and differences among training activities.

### 4.2   ACTIVITY SPLITTING

We propose activity splitting to divide the all-in-one activity $\alpha_0$ into multiple groups after it is trained for certain rounds. Essentially, we aim to split $\mathcal{A} = \{\alpha_1, \alpha_2, \ldots, \alpha_n\}$ into multiple nonoverlapping groups such that training activities within a group have better synergy. Let $\{\mathcal{A}_1, \mathcal{A}_2, \ldots, \mathcal{A}_m\}$ be subsets of $\mathcal{A}$, we aim to find a disjoint set $I$ of $\mathcal{A}$, where $I \subseteq \{1, 2, \ldots, m\}, |I| \leq |\mathcal{A}|, \bigcup_{j \in I} \mathcal{A}_j = \mathcal{A}$, and $\bigcap_{j \in I} \mathcal{A}_j = \emptyset$. Each group $\mathcal{A}_j$ trains a model $\nu_j = \{\theta_s^j\} \cup \{\theta_{\alpha_i} | \alpha_i \in \mathcal{A}_j\}$, which is a multi-task network when $\mathcal{A}_j$ contains more than one training activity, where $\theta_s^j$ is the shared model parameters and $\theta_{\alpha_i}$ is the specific parameters for training activity $\alpha_i \in \mathcal{A}_j$. The core question is how to determine set $I$ to split these activities considering their synergies and differences.

Inspired by TAG (Fifty et al., 2021) that measures task affinites for task grouping, we employ affinities between training activities for activity splitting via three stages: 1) we measure affinities among activities during *all-in-one* training; 2) we select the best combination of splitted training activities based on affinity scores; 3) we continue training each split with its model initialized with parameters obtained from all-in-one training. Particularly, during training of all-in-one activity $\alpha_0$, we measure the affinity of training activity $\alpha_i$ onto $\alpha_j$ at time step $t$ in each client $k$ with the following equation:

$$\mathcal{S}_{\alpha_i \to \alpha_j}^{k,t} = 1 - \frac{\mathcal{L}_{\alpha_j}(\mathcal{X}^{k,t}, \theta_{s,\alpha_i}^{k,t+1}, \theta_{\alpha_j}^{k,t})}{\mathcal{L}_{\alpha_j}(\mathcal{X}^{k,t}, \theta_s^{k,t}, \theta_{\alpha_j}^{k,t})}, \tag{2}$$

where $\mathcal{L}_{\alpha_j}$ is the loss function of $\alpha_j$, $\mathcal{X}^{k,t}$ is a batch of training data, and $\theta_s^{k,t}$ and $\theta_{s,\alpha_i}^{k,t+1}$ are the shared model parameters *before* and *after* updated by $\alpha_i$, respectively. Positive value of $\mathcal{S}_{\alpha_i \to \alpha_j}^{k,t}$

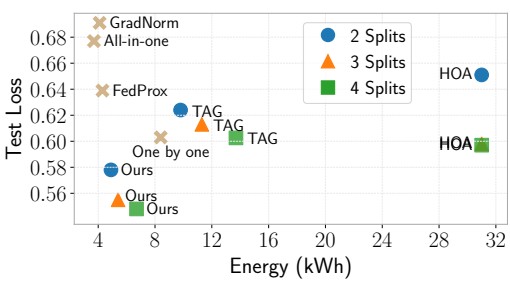 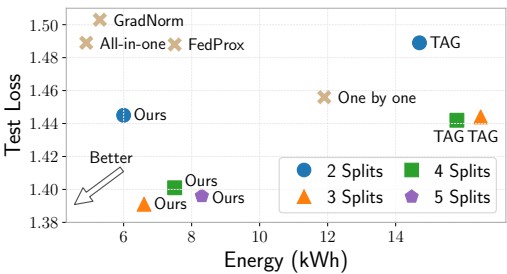

(a) A set of five training activities: `sdnkt`        (b) A set of nine trianing activities: `sdnkterca`

Figure 2: Comparison of test loss and energy consumption on two training activity sets: (a) `sdnkt` and (b) `sdnkterca`, where each character represents an activity. Compared with all-in-one methods, our method achieves much better performance with slight increases in computation. Moreover, our method achieves the best performance while consuming less energy than the other methods.

means that activity $\alpha_i$ helps reduce the loss of $\alpha_j$. This equation measures the affinity of one time-step of one client. We approximate affinity scores for each round by averaging the values over $T$ time-steps in $E$ local epochs and $K$ selected clients: $\hat{\mathcal{S}}_{\alpha_i \to \alpha_j} = \frac{1}{KET} \sum_{k=1}^{K} \sum_{e=1}^{E} \sum_{t=1}^{T} \mathcal{S}_{\alpha_i \to \alpha_j}^{k,t}$, where $T$ is total time steps determined by the frequency $f$ of calculating Equation 2, e.g., $f = 5$ means measuring the affinity in each client in every five batches.

These affinity scores measure pair-wise affinities between traininig activities. We next use them to calculate total affinity scores of a grouping with $\sum_{i=1}^{n} \hat{\mathcal{S}}_{\alpha_i}$, where $\hat{\mathcal{S}}_{\alpha_i}$ is the averaged affinity score onto each training activity. For example, in a grouping of two splits of five training activities $\{\{\alpha_1, \alpha_2\}, \{\alpha_3, \alpha_4, \alpha_5\}\}$, where $\{\alpha_1, \alpha_2\}$ is one split and $\{\alpha_3, \alpha_4, \alpha_5\}$ is another split. The affinity score onto $\alpha_1$ is $\hat{\mathcal{S}}_{\alpha_1} = \hat{\mathcal{S}}_{\alpha_2 \to \alpha_1}$ and the affinity score onto $\alpha_3$ is $\hat{\mathcal{S}}_{\alpha_3} = (\hat{\mathcal{S}}_{\alpha_4 \to \alpha_3} + \hat{\mathcal{S}}_{\alpha_5 \to \alpha_3})/2$. Consequently, we can find the set $I$ with $|I|$ elements for subsets of $\mathcal{A}$ that maximize $\sum_{i=1}^{n} \hat{\mathcal{S}}_{\alpha_i}$, where $|I|$ defines the number of elements.

We would like to further highlight the differences between our method and TAG (Fifty et al., 2021). Firstly, TAG focuses on inference efficiency, thus it allows overlapping task grouping that could train one task multiple times. In contrast, our focus is fundamentally different; we focus on training efficiency and consider only nonoverlapping activity splitting. Secondly, TAG is computation-intensive for higher numbers of splits, e.g., it fails to produce results of five splits of nine tasks in a week, whereas we only need seconds of computation. Thirdly, TAG rules out the possibility that a group contains only one task as it sets $\hat{\mathcal{S}}_{\alpha_i \to \alpha_i} = 1e^{-6}$, which is much smaller than scores of other groupings. Besides, $\hat{\mathcal{S}}_{\alpha_i \to \alpha_i}$ calculated from Equation 2 is also not desirable; it always results in a group containing only one task as its value is much larger (could be 10x larger) than scores of other groupings. To overcome these issues, we propose a new method to calculate this value:

$$\hat{\mathcal{S}}_{\alpha_i \to \alpha_i} = \sum_{j \in \mathcal{N} \setminus \{i\}} \frac{(\hat{\mathcal{S}}_{\alpha_i \to \alpha_j} + \hat{\mathcal{S}}_{\alpha_j \to \alpha_i})}{2n - 2}, \tag{3}$$

where $\mathcal{N} = \{1, 2, \ldots, n\}$. The intuition is that it measures the normalized affinity of activity $\alpha_i$ to other activities and other activities to $\alpha_i$. Fourthly, we focus on multi-tenant FL, thus, we further aggregate affinity scores over $K$ selected clients. Fifthly, TAG trains each set $\mathcal{A}_j$ from scratch, whereas we initialize their models with the parameters obtained from all-in-one training.

## 5 EXPERIMENTS

We evaluate the performance and resource usage of MuFL and design our experiments to answer the following questions: 1) How effective is our activity splitting approach? 2) When to split the training activities? 3) Is it beneficial to iteratively split the training activities? 4) What is the impact of local epoch and scaling up the number of selected clients in each training round?

Table 1: Performance (test loss) comparison of our method with the optimal and worst splits. Our method achieves the best performance, indicating the effectiveness of our activity splitting method.

| Activity Set | Splits | Ours | Train from Scratch | | Train from Initialization | |
|---|---|---|---|---|---|---|
| | | | Optimal | Worst | Optimal | Worst |
| sdnkt | 2 | **0.578 ± 0.015** | 0.622 ± 0.007 | 0.685 ± 0.010 | 0.595 ± 0.008 | 0.595 ± 0.004 |
| | 3 | **0.555 ± 0.008** | 0.585 ± 0.026 | 0.674 ± 0.022 | 0.560 ± 0.006 | 0.578 ± 0.006 |
| erckt | 2 | **1.039 ± 0.024** | 1.070 ± 0.013 | 1.312 ± 0.065 | 1.048 ± 0.024 | 1.068 ± 0.037 |
| | 3 | **1.015 ± 0.018** | 1.058 ± 0.029 | 1.243 ± 0.099 | 1.020 ± 0.012 | 1.052 ± 0.026 |

**Experiment Setup** We construct the *Scenario 1* of multi-tenant FL scenarios using Taskonomy dataset (Zamir et al., 2018), which is a large computer vision dataset of indoor scenes of buildings. We run experiments with $N = 32$ clients, where each client contains a dataset of a building to simulate the statistical heterogeneity in FL. Three sets of training activities are used to evaluate the robustness of MuFL: sdnkt, erckt, and sdnkterca; each character represents an activity, e.g., s represents semantic segmentation. We measure the statistical performance of an activity set using the sum of test losses of individual activities. By default, we use $K = 4$ selected clients and $E = 1$ local epoch for each round of training. More experimental details are provided in Appendix B.

## 5.1 Performance Evaluation

We compare the performance, in terms of test loss and energy consumption, among the following methods: 1) one by one training of activities (i.e., the vanilla multi-tenant FL); 2) all-in-one training of activities (i.e., using only activity consolidation); 3) all-in-one training with multi-task optimization (GradNorm (Chen et al., 2018)) and federated optimization (FedProx (Li et al., 2020)); 4) estimating higher-order of activity groupings from pair-wise activities performance (HOA (Standley et al., 2020)); 5) grouping training activities with only task affinity grouping method (TAG (Fifty et al., 2021)); 6) MuFL with both activity consolidation and activity splitting. Carbontracker (Anthony et al., 2020) is used to measure energy consumption and carbon footprint (provided in Appendix C).

Figure 2 compares performance of the above methods on activity sets sdnkt and sdnkterca. The methods that achieve lower test loss and lower energy consumption are better. At the one extreme, all-in-one methods (including GradNorm) consumes the least energy, but their test losses are the highest. Simply applying federated optimization, FedProx, can hardly improve performance, especially on sdnkterca. At the other extreme, HOA achieves comparable test losses on three or four splits of sdnkt, but it demands high energy consumption ($\sim 4 - 6\times$ of ours) to compute pair-wise activities for higher-order estimation. Although training activities one by one and TAG present a good balance between test loss and energy consumption, MuFL is superior in both aspects; it achieves the best test loss with $\sim40\%$ and $\sim50\%$ less energy consumption on activity set sdnkt and sdnkterca, respectively. Additionally, more splits of activity in the activity splitting lead to higher energy consumption, but it could help further reduce test losses. We do not report HOA for activity set sdnkterca due to computation constraints.[2] We omit to report the running time as it is hidden under the metric of energy consumption; higher energy consumption implies longer training time. We provide more details of these experiments and results of activity set erckt in Appendix C.

## 5.2 How Effective is Our Activity Splitting Approach?

We demonstrate the effectiveness of our activity splitting approach by comparing it with the possible optimal and worst splits. The optimal and worst splits are obtained with two steps: 1) we measure the performance over all combinations of two splits and three splits of an activity set by training them from scratch;[3] 2) we select the combination that yields the best performance as the optimal split and the worst performance as the worst split.

Table 1 compares the test loss of MuFL with the optimal and worst splits trained in two ways: 1) training each split from scratch; 2) training each split the same way as our activity splitting — initializing models with the parameters obtained from all-in-one training. On the one hand, training

---

[2]HOA computes at least 36 pairs of activities ($\sim$720 GPU hours), consuming $\sim12\times$ more energy than MuFL.

[3]There are fifteen and twenty-five combinations of two and three splits, respectively, for a set of five activities.

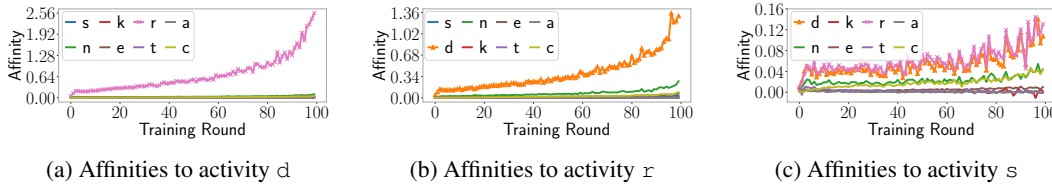

(a) Affinities to activity d (b) Affinities to activity r (c) Affinities to activity s

Figure 3: Changes of affinity scores of one activity to the other on activity set sdnkterca. Activities d and r have high inter-activity scores. The trends of affinities emerge at the early stage of training.

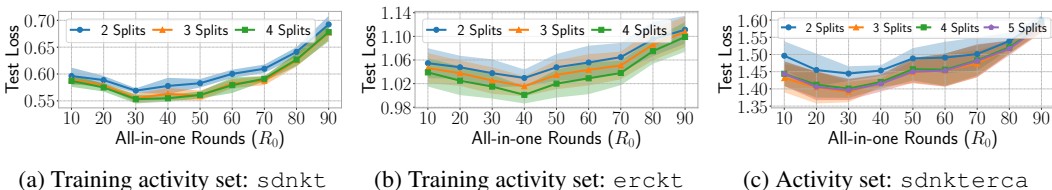

(a) Training activity set: sdnkt (b) Training activity set: erckt (c) Activity set: sdnkterca

Figure 4: Performance comparison of training all-in-one activities for different $R_0$ rounds. Fixing the total training rounds $R = 100$, our method achieves the best performance when $R_0 \in \{20, 30, 40\}$.

from initialization outperforms training from scratch in all settings. It suggests that initializing each split with all-in-one training model parameters can significantly improve the performance. On the other hand, our activity splitting method achieves the best performance in all settings, even though training from initialization reduces the gaps of different splits (the optimal and worst splits). These results indicate the effectiveness of our activity splitting approach.

## 5.3 WHEN TO SPLIT TRAINING ACTIVITIES?

We further answer the question that how many $R_0$ rounds should we train the all-in-one activity before activity splitting. It is determined by two factors: 1) the rounds needed to obtain affinity scores for a reasonable activity splitting; 2) the rounds that yield the best overall performance.

**Affinity Analysis** We analyze changes in affinity scores over the course of training to show that early-stage affinity scores are acceptable for activity splitting. Figure 3 presents the affinity scores of different activities to one activity on activity set sdnkterca. Figure 3a and 3b indicate that activity d and activity r have high inter-activity affinity scores; they are divided into the same group as a result. In contrast, both d and r have high affinity score to activity s in Figure 3c, but not vice versa. These trends emerge in the early stage of training, thus, we employ the affinity scores of the *tenth* round for activity splitting by default; they are effective in achieving promising results as shown in Figure 2 and Table 1. We provide more affinity scores of other activities in Appendix C.

**The Impact of $R_0$ Rounds** Figure 4 compares the performance of training $R_0$ for 10 to 90 rounds before activity splitting. Fixing the total training round $R = 100$, we train each split of activities for $R_1 = R - R_0$ rounds. The results indicate that MuFL achieves the best performance when $R_0 = \{20, 30, 40\}$ rounds. Training the all-in-one activity for enough rounds helps utilize the benefits and synergies of training together, but training for too many rounds almost suppresses the benefits of considering differences among activities. We suggest training $R_0$ for [20, 40] that strikes a good balance between these two extremes and consider other mechanisms to determine $R_0$ in future works.

## 5.4 HIERARCHICAL SPLITTING

This section evaluates an alternative activity splitting strategy. In activity splitting, we can divide the all-in-one training activity into $\{2, 3, \dots\}$ splits. As shown in Figure 2, more splits lead to better performance with slightly higher energy consumption in the five-activity set, but the trend is not straightforward in the nine-activity set. Apart from setting the number of splits directly, MuFL can split the training activity into more splits adaptively via two steps: 1) dividing the all-in-one activity into two splits and training each one for $R_1$ rounds; 2) further dividing one of them to two splits and train these three activities for $R_2$ rounds. We term the adaptive process as *hierarchical splitting*.

Table 2: Performance of hierarchical splitting on three activity sets `sdnkt`, `erckt`, and `sdnkterca`. Hierarchical splitting outperforms two splits and achieves similar performance to three splits with less energy (kWh) consumption.

| Method | sdnkt | | erckt | | sdnkterca | |
|---|---|---|---|---|---|---|
| | Energy | Test Loss | Energy | Test Loss | Energy | Test Loss |
| Two Splits | $4.9 \pm 0.3$ | $0.578 \pm 0.015$ | $6.7 \pm 0.2$ | $1.039 \pm 0.024$ | $6.0 \pm 0.1$ | $1.445 \pm 0.021$ |
| Three Splits | $5.4 \pm 0.7$ | $0.555 \pm 0.008$ | $7.2 \pm 0.2$ | $1.015 \pm 0.018$ | $6.6 \pm 0.4$ | $1.391 \pm 0.030$ |
| Hierarchical | $\mathbf{5.3 \pm 0.4}$ | $0.563 \pm 0.007$ | $\mathbf{6.9 \pm 0.2}$ | $1.022 \pm 0.020$ | $\mathbf{6.5 \pm 0.3}$ | $1.403 \pm 0.024$ |

Table 2 compares the performance of hierarchical splitting (3 splits) with directly splitting to multiple splits on three activity sets. We use $R_0 = 30$, $R_1 = 40$, and $R_2 = 30$ for activity sets `sdnkt` and `erckt`, and $R_0 = 30$, $R_1 = 20$, and $R_2 = 50$ for `sdnkterca`. Hierarchical splitting effectively reduces test losses of two splits and achieves comparable performance to three splits with less energy consumption. These results suggest that hierarchical splitting can be an alternative method of activity splitting. Additionally, these results also demonstrate the possibility of other activity splitting strategies to be considered in future works.

## 5.5 ADDITIONAL ANALYSIS

This section analyzes the impact of local epoch $E$ and the number of selected clients $K$ in FL using all-in-one training. We report the results of activity set `sdnkt` here and provide more results in Appendix C.

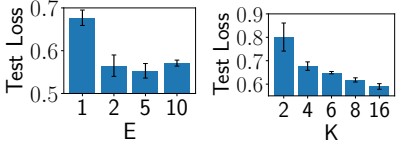

(a) Impact of $E$      (b) Impact of $K$

**Impact of Local Epoch $E$** Local epoch defines the number of epochs each client trains before uploading training updates to the server. Figure 5a compares test losses of local epochs $E = \{1, 2, 5, 10\}$. Larger $E$ could lead to better performance with higher computation (fixed training round $R = 100$), but it is not effective when increasing $E = 5$ to $E = 10$. It suggests the limitation of simply increasing computation with larger $E$ in improving performance. Note that MuFL (Table 1) achieves better results than $E = 5$ with $\sim 5\times$ less computation.

Figure 5: Analysis of the impact of (a) local epoch $E$ and (b) the number of selected clients $K$ on activity set `sdnkt`. Larger $E$ and $K$ could reduce losses with more computation, but the benefit decreases as computation increases.

**Impact of The Number of Selected Clients $K$** Figure 5b compares test losses of the number of selected clients $K = \{2, 4, 6, 8, 16\}$ in each round. Increasing the number of selected clients improves the performance, but the effect becomes marginal as $K$ increases. Larger $K$ can also be considered as using more computation in each round. Similar to the results of the impact of $E$, simply increasing computation can only improve performance to a certain extent. It also shows the significance of MuFL that increases performance with slightly more computation. We use $K = 4$ by default for experiments and demonstrate that MuFL is also effective on $K = 8$ in Appendix C.

## 6 CONCLUSIONS

In this work, we propose a smart multi-tenant federated learning system to effectively coordinate and execute multiple simultaneous FL training activities. In particular, we introduce activity consolidation and activity splitting to consider both synergies and differences among training activities. Extensive empirical studies demonstrate that our method is effective in elevating performance and significant in reducing energy consumption and carbon footprint by more than 40%, which are important metrics to our society. We believe that multi-tenant FL will emerge and empower many real-world applications with the fast development of FL. We hope this research will inspire the community to further work on algorithm and system optimizations of multi-tenant FL. Future work involves designing better scheduling mechanisms to coordinate training activities. Client selection strategies can also be considered to optimize resource and training allocation, and extend our optimization approaches to other multi-tenant FL scenarios.

## 7 REPRODUCIBILITY STATEMENT

To facilitate reproducibility, we provide basic experimental setups in Section 5 and include more details about the dataset, implementation details, and hyperparameters in Appendix B. We also provide the algorithm of MuFL in Algorithm 1. Besides, the implementation codes will be open-sourced in the future.

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

## A   MULTI-TENANT FL SCENARIOS

We introduce four multi-tenant federated learning scenarios in Section 3. Figure 6 depicts these four scenarios with variances in two aspects: 1) whether all training activities are the same type of application, e.g., CV applications; 2) whether all clients support all training activities.

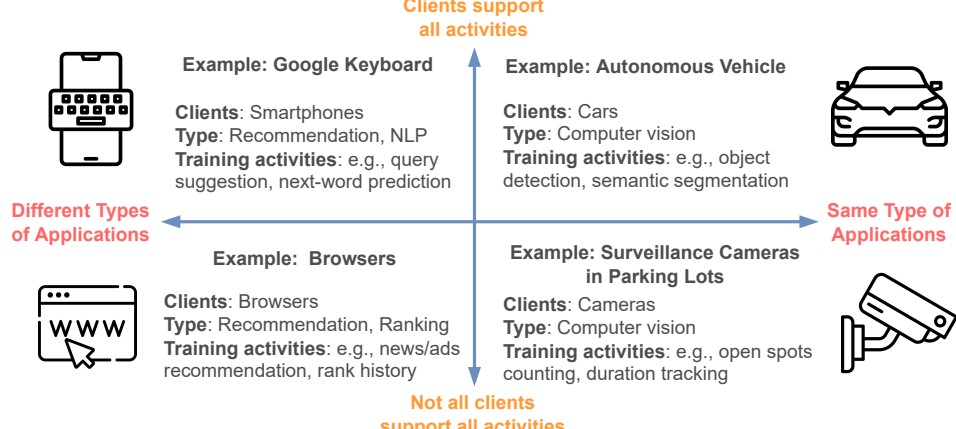

Figure 6: Illustration of the four multi-tenant FL scenarios.

Our proposed approach, MuFL, focuses on optimizing the performance on Scenario 1. Optimizing Scenario 1 can potentially empower plenty of real-world applications. For example, autonomous vehicles relate to multiple computer vision (CV) tasks (Janai et al., 2020), including object detection, tracking, and semantic segmentation; smart city surveillance cameras associate with various CV tasks, such as crowd counting, object detection, and person re-identification (Zhuang et al., 2020); voice assistant applications like Apple Siri and Google Assistant need multiple automatic speech recognition (ASR) tasks, including word confidence, word deletion, and utterance confidence (Qiu et al., 2021); household robots like Amazon Astro need to perform multiple CV tasks such as visual odometry tracking, loop-closure detection, and object detection (Ye & Sen, 2022); smart-manufacturing robots need several CV tasks such as object detection, object grasp point detection, and object pose estimation (FLAIROP, 2022).

## B   EXPERIMENTAL DETAILS

This section provides more experimental information, including dataset, implementation details, and computation resources used.

**Dataset**   We run experiments using Taskonomy dataset (Zamir et al., 2018), which is a large computer vision (CV) dataset of indoor scenes of buildings. To facilitate reproducibility and mitigate computational requirements, we use the tiny split of Taskonomy dataset,[4] whose size is around 445GB. We select nine CV applications to form three sets of training activities: `sdnkt`, `erckt`, `sdnkterca`. These nine actvities are also used in (Standley et al., 2020). Figure 8 provides sample images of these nine training activities, as well as the representation of each character.[5] In particular, we employ indoor images

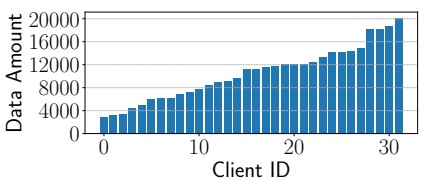

Figure 7: The data amount distribution of each training activity over 32 clients.

---

[4]Taskonomy dataset is released under MIT license and can be downloaded from their official repository https://github.com/StanfordVL/taskonomy.

[5]The meaning of each character in `sdnkterca` are as follows; `s`: semantic segmentation, `d`: depth estimation, `n`: normals, `k`: keypoint, `t`: edge texture, `e`: edge occlusion, `r`: reshaping, `c`: principle curvature, `a`: auto-encoder.

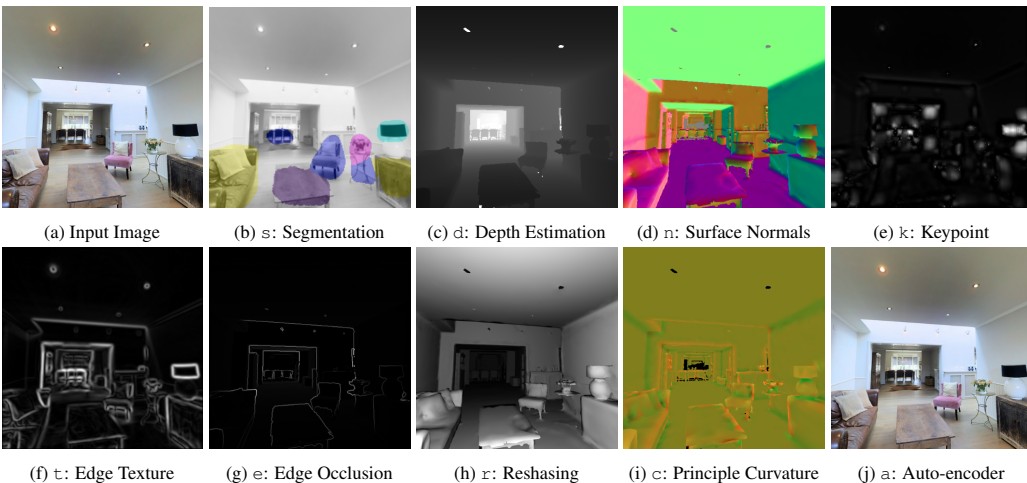

Figure 8: Sample images of nine training activities corresponding to the input image.

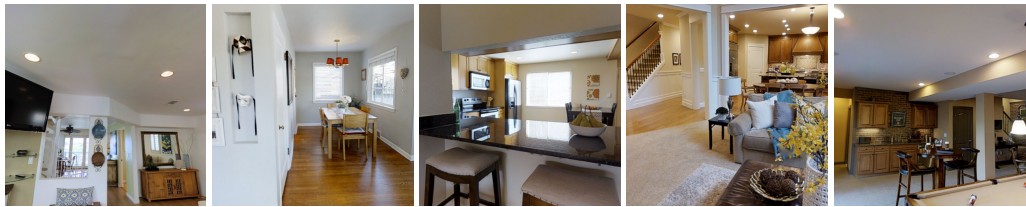

Figure 9: Sample images of five clients, where each client contains indoor scenes of a building. These indoor images differ in design, layout, objects, and illumination.

of 32 buildings [6] as the total number of clients $N = 32$; each client contains images of a building to simulate the statistical heterogeneity. On the one hand, clients have different sizes of data. Figure 7 shows the distribution of dataset sizes of an activity of clients. On the other hand, Figure 9 shows sample images of five clients; their indoor scenes vary in design, layout, objects, and illumination.

**Implementation Details** We implement multi-tenant FL systems in Python using PyTorch (Paszke et al., 2017). We simulate the FL training on a cluster of NVIDIA Tesla V100 GPUs, where each node in the cluster contains 8 GPUs. In each round, each selected client is allocated to a GPU to conduct training; these clients communicate via the NCCL backend. Besides, we employ FedAvg (McMahan et al., 2017) for the server aggregation. By default, we randomly select $K = 4$ clients to train for $E = 1$ local epochs in each round and train for $R = 100$ rounds.

We reference the implementation of multi-task learning from (Standley et al., 2020)'s official repository [7] for all-in-one training and training of each split after activity splitting. Particularly, the network architecture contains an encoder $\theta_s$ and multiple decoders $\theta_{\alpha_i}$; one decoder for a training activity $\alpha_i$. Figure 10 illustrates the network architectures of training activities before and after activity consolidation and activity splitting. We use the modified Xception Network (Chollet, 2017) as the encoder for activity sets `sdnkt` and `erckt` and half size of the network (half amount of parameters) for activity set `sdnkterca`. The decoders contain four deconvolution layers and four convolution layers. Each training activity contains a loss function. Specifically, semnatic segmentation `s` uses Cross Entropy loss; surface normals and depth estimation use rotation loss based on L1 loss; keypoint detection, edge occlusion, edge texture, auto encoder, and principle curvature use L1 loss. We refer

---

[6]The name of the buildings are allensville, beechwood, benevolence, coffeen, collierville, corozal, cosmos, darden, forkland, hanson, hiteman, ihlen, klickitat, lakeville, leonardo, lindenwood, markleeville, marstons, mcdade, merom, mifflinburg, muleshoe, newfields, noxapater, onaga, pinesdale, pomaria, ranchester, shelbyville, stockman, tolstoy, and uvalda.

[7]https://github.com/tstandley/taskgrouping

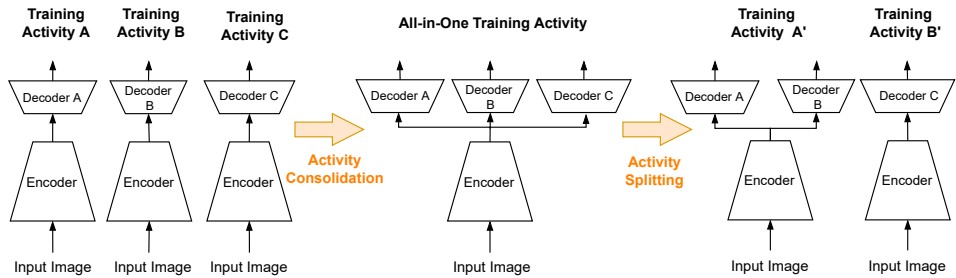

Figure 10: Illustration of network architectures of training activities in MuFL. Initially, each activity employs an encoder and a decoder. Activity consolidation consolidates these training activities into an all-in-one activity with multi-task architecture. After activity splitting, MuFL divides the all-in-one activity into multiple training activities, each contains an encoder and one or multiple decoders.

implementation of loss functions from (Standley et al., 2020) [8]. The batch size is $B = 64$ for `sdnkt` and `erckt` and $B = 32$ for `sdnkterca`. These are the maximum batch sizes for one GPU without out-of-memory issues. In addition, we use polynomial learning rate decay $(1 - \frac{r}{R})^{0.9}$ to update learning rate in each round with initial learning rate $\eta = 0.1$, where $r$ is the number of trained rounds and $R = 100$ is the default total training rounds. The optimizer is stochastic gradient descent (SGD), with momentum of 0.9 and weight decay $1e^{-4}$.

**Implementation of Compared Methods**    We tune the hyperparameter $\mu = 0.004$ for the proximal term in FedProx (Li et al., 2020). GradNorm (Chen et al., 2018) implementation is adopted from (Standley et al., 2020; Fifty et al., 2021) with default $\alpha = 1.5$ and TAG (Fifty et al., 2021) implementation is adopted from their official repository [9]. Next, we provide the details of how we compute the results of HOA (Standley et al., 2020) and TAG (Fifty et al., 2021).

HOA (Standley et al., 2020) needs to compute test losses for individual activities and pair-wise activity combinations for $R = 100$ rounds. After that, we use these results to estimate test losses of higher-order combinations following (Standley et al., 2020). We then compute the actual test losses for the optimal activity splits that have the lowest test losses by training them from scratch. For example, for activity set `sdnkt`, we compute s, d, n, k, t and ten pair-wise activity combinations. Then, we use these results to estimate test losses of higher-order combinations.

TAG (Fifty et al., 2021) first computes all-in-one training for $R = 100$ rounds to obtain the pair-wise affinities. Then, it uses a network selection algorithm to group these activities. After that, we train each group of activities from scratch for $R = 100$ rounds to obtain test losses. The best result is reported for overlapping activities. For example, {sd, dn, kt} is the best result of three splits of TAG on activity set `sdnkt`. Then, each split is trained from scratch to obtain test losses.

**Computation Resources**    Experiments in this work take approximately 27,765 GPU hours of NVIDIA Tesla V100 GPU for training. We conduct three independent runs of experiments for the majority of empirical studies. In each run, activity set `sdnkt` takes around 2,330 GPU hours, `erckt` takes around 3,280 GPU hours, and `sdnkterca` takes around 3,645 GPU hours. These include experiments of compared methods and ablation studies, whereas these do not include the GPU hours for validation and testing. It takes around the same GPU hours as training when we validate the model after each training round.

## C    ADDITIONAL EXPERIMENTAL EVALUATION

This section provides more experimental results, including comprehensive results of performance evaluation and additional ablation studies.

---

[8]https://github.com/tstandley/taskgrouping/blob/master/taskonomy_losses.py

[9]https://github.com/google-research/google-research/tree/master/tag

Table 3: Comparison of test loss, energy consumption, and carbon footprint on activity set `sdnkt`.

| Method | Splits | Energy (kWh) | CO2eq (g) | Total Loss | s | d | n | k | t |
|---|---|---|---|---|---|---|---|---|---|
| One by one | - | 8.4 ± 0.1 | 2465 ± 39 | 0.603 ± 0.030 | 0.086 ± 0.005 | 0.261 ± 0.023 | 0.107 ± 0.001 | 0.107 ± 0.003 | 0.043 ± 0.002 |
| All-in-one | - | 3.7 ± 0.1 | 1086 ± 28 | 0.677 ± 0.018 | 0.087 ± 0.002 | 0.246 ± 0.010 | 0.136 ± 0.001 | 0.126 ± 0.019 | 0.083 ± 0.008 |
| GradNorm | - | 4.1 ± 0.4 | 1200 ± 122 | 0.691 ± 0.013 | 0.092 ± 0.001 | 0.251 ± 0.012 | 0.138 ± 0.003 | 0.118 ± 0.007 | 0.093 ± 0.019 |
| HOA | 2 | 31.0 ± 0.5 | 9125 ± 140 | 0.651 ± 0.029 | 0.091 ± 0.011 | 0.245 ± 0.002 | 0.135 ± 0.000 | 0.107 ± 0.003 | 0.074 ± 0.023 |
| TAG | 2 | 9.8 ± 0.3 | 2876 ± 88 | 0.624 ± 0.015 | 0.083 ± 0.004 | 0.242 ± 0.005 | 0.134 ± 0.001 | 0.110 ± 0.007 | 0.055 ± 0.006 |
| **MuFL** | 2 | 4.9 ± 0.3 | 1431 ± 94 | 0.578 ± 0.015 | 0.069 ± 0.006 | 0.231 ± 0.006 | 0.124 ± 0.002 | 0.102 ± 0.003 | 0.052 ± 0.003 |
| HOA | 3 | 31.0 ± 0.5 | 9125 ± 140 | 0.598 ± 0.029 | 0.083 ± 0.022 | 0.239 ± 0.007 | 0.127 ± 0.008 | 0.107 ± 0.003 | 0.043 ± 0.002 |
| TAG | 3 | 11.3 ± 0.2 | 3313 ± 56 | 0.613 ± 0.032 | 0.094 ± 0.005 | 0.233 ± 0.002 | 0.122 ± 0.013 | 0.110 ± 0.008 | 0.055 ± 0.008 |
| **MuFL** | 3 | 5.4 ± 0.3 | 1589 ± 94 | 0.555 ± 0.015 | 0.072 ± 0.006 | 0.222 ± 0.006 | 0.124 ± 0.002 | 0.095 ± 0.003 | 0.042 ± 0.003 |
| HOA | 4 | 31.0 ± 0.5 | 9125 ± 140 | 0.597 ± 0.015 | 0.094 ± 0.009 | 0.238 ± 0.002 | 0.115 ± 0.014 | 0.107 ± 0.003 | 0.043 ± 0.002 |
| TAG | 4 | 13.7 ± 0.3 | 4016 ± 80 | 0.603 ± 0.027 | 0.083 ± 0.005 | 0.233 ± 0.002 | 0.122 ± 0.013 | 0.110 ± 0.008 | 0.055 ± 0.008 |
| **MuFL** | 4 | 6.7 ± 0.3 | 1969 ± 75 | 0.548 ± 0.001 | 0.070 ± 0.002 | 0.230 ± 0.008 | 0.111 ± 0.000 | 0.095 ± 0.007 | 0.042 ± 0.001 |

Table 4: Comparison of test loss, energy consumption, and carbon footprint on activity set `erckt`.

| Method | Splits | Energy (kWh) | CO2eq (g) | Total Loss | e | r | c | k | t |
|---|---|---|---|---|---|---|---|---|---|
| One by one | - | 11.1 ± 2.2 | 3277 ± 660 | 1.055 ± 0.034 | 0.148 ± 0.000 | 0.371 ± 0.029 | 0.386 ± 0.006 | 0.107 ± 0.003 | 0.043 ± 0.002 |
| All-in-one | - | 5.0 ± 0.3 | 1478 ± 84 | 1.130 ± 0.022 | 0.146 ± 0.001 | 0.379 ± 0.019 | 0.393 ± 0.002 | 0.110 ± 0.003 | 0.079 ± 0.013 |
| GradNorm | - | 5.0 ± 0.2 | 1462 ± 70 | 1.154 ± 0.055 | 0.147 ± 0.002 | 0.381 ± 0.015 | 0.394 ± 0.001 | 0.149 ± 0.062 | 0.082 ± 0.005 |
| HOA | 2 | 38.3 ± 0.3 | 11265 ± 86 | 1.082 ± 0.032 | 0.149 ± 0.003 | 0.365 ± 0.025 | 0.394 ± 0.002 | 0.109 ± 0.002 | 0.064 ± 0.022 |
| TAG | 2 | 14.0 ± 0.9 | 4119 ± 279 | 1.095 ± 0.033 | 0.147 ± 0.002 | 0.379 ± 0.013 | 0.393 ± 0.000 | 0.108 ± 0.005 | 0.068 ± 0.015 |
| **MuFL** | 2 | 6.7 ± 0.2 | 1957 ± 53 | 1.039 ± 0.024 | 0.143 ± 0.001 | 0.343 ± 0.014 | 0.393 ± 0.001 | 0.104 ± 0.006 | 0.056 ± 0.007 |
| HOA | 3 | 38.3 ± 0.2 | 11265 ± 53 | 1.062 ± 0.024 | 0.149 ± 0.001 | 0.365 ± 0.014 | 0.394 ± 0.001 | 0.109 ± 0.006 | 0.046 ± 0.007 |
| TAG | 3 | 14.4 ± 0.6 | 4242 ± 170 | 1.091 ± 0.034 | 0.147 ± 0.002 | 0.388 ± 0.014 | 0.396 ± 0.002 | 0.109 ± 0.009 | 0.050 ± 0.011 |
| **MuFL** | 3 | 7.2 ± 0.2 | 2108 ± 50 | 1.015 ± 0.018 | 0.143 ± 0.000 | 0.336 ± 0.005 | 0.383 ± 0.001 | 0.102 ± 0.008 | 0.052 ± 0.009 |
| HOA | 4 | 38.3 ± 0.3 | 11265 ± 86 | 1.053 ± 0.034 | 0.148 ± 0.002 | 0.369 ± 0.028 | 0.386 ± 0.006 | 0.105 ± 0.001 | 0.045 ± 0.003 |
| TAG | 4 | 17.4 ± 0.5 | 5114 ± 159 | 1.087 ± 0.028 | 0.147 ± 0.002 | 0.384 ± 0.011 | 0.396 ± 0.002 | 0.109 ± 0.009 | 0.050 ± 0.011 |
| **MuFL** | 4 | 7.6 ± 0.0 | 2229 ± 14 | 1.002 ± 0.014 | 0.143 ± 0.000 | 0.336 ± 0.005 | 0.383 ± 0.001 | 0.094 ± 0.009 | 0.046 ± 0.004 |

## C.1 PERFORMANCE EVALUATION

Table 3 and 5 provide comprehensive comparison of different methods on test loss and energy consumption on activity sets `sdnkt` and `sdnkterca`, respectively. They complement the results in Figure 2. Besides, Table 4 and Figure 11 compares these methods on activity set `erckt`. The results on `erckt` is similar to results on the other activity sets; our method achieves the best performance with around 40% less energy consumption than the one-by-one method and with slightly more energy consumption than all-in-one methods.

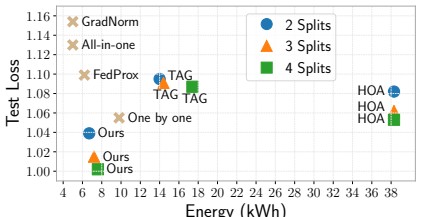

Figure 11: Compare test loss and energy consumption on activity set `erckt`.

Additionally, Table 3, 4 and 5 also provide carbon footprints (CO2eq) of different methods. The carbon footprints are estimated using Carbontracker (Anthony et al., 2020).[10] Our method reduces around 40% on carbon footprints on these three activity sets compared with one-by-one training; it reduces 1526gCO$_2$eq or equivalent to traveling 12.68km by car on `sdnkterca`. The reduction is even more significant when compared with TAG and HOA. Although we run experiments using Tesla V100 GPU, the relative results of energy and carbon footprint among different methods should be representative of the scenarios of edge devices.

## C.2 ADDITIONAL ANALYSIS AND ABLATION STUDIES

This section presents additional analysis of MuFL and provides additional ablation studies.

---

[10]Carbon intensity of a training varies over geographical regions according to (Anthony et al., 2020). We use the national level (the United Kingdom as the default setting of the tool) of carbon intensity for a fair comparison across different methods. These carbon footprints serve as a proxy for evaluation of the actual carbon emissions.

Table 5: Comparison of test loss, energy consumption, and carbon footprint on `sdnkterca`.

| Method | Splits | Energy | CO2eq (g) | Total Loss | s | d | n | k | t | e | r | c | a |
|---|---|---|---|---|---|---|---|---|---|---|---|---|---|
| One by one | - | 11.9±0.5 | 3512±151 | 1.46±0.011 | 0.08±0.009 | 0.24±0.014 | 0.10±0.001 | 0.10±0.002 | 0.04±0.003 | 0.15±0.001 | 0.35±0.011 | 0.38±0.002 | 0.02±0.000 |
| All-in-one | - | 4.9±0.2 | 1435±60 | 1.49±0.025 | 0.09±0.002 | 0.23±0.009 | 0.13±0.002 | 0.10±0.002 | 0.07±0.005 | 0.14±0.001 | 0.33±0.011 | 0.39±0.001 | 0.02±0.001 |
| GradNorm | - | 5.3±1.3 | 1561±377 | 1.50±0.049 | 0.08±0.004 | 0.24±0.014 | 0.13±0.003 | 0.10±0.003 | 0.07±0.011 | 0.14±0.001 | 0.34±0.018 | 0.39±0.001 | 0.02±0.001 |
| TAG | 2 | 14.7±0.8 | 4317±229 | 1.49±0.025 | 0.09±0.002 | 0.23±0.008 | 0.13±0.002 | 0.10±0.002 | 0.07±0.005 | 0.14±0.001 | 0.33±0.011 | 0.39±0.001 | 0.02±0.001 |
| MuFL | 2 | 6.0±0.1 | 1986±108 | 1.45±0.021 | 0.08±0.003 | 0.22±0.008 | 0.12±0.001 | 0.10±0.001 | 0.06±0.004 | 0.14±0.000 | 0.32±0.011 | 0.39±0.001 | 0.02±0.001 |
| TAG | 3 | 16.5±2.6 | 4854±751 | 1.44±0.014 | 0.09±0.006 | 0.23±0.009 | 0.12±0.001 | 0.10±0.002 | 0.03±0.004 | 0.14±0.000 | 0.33±0.009 | 0.39±0.001 | 0.02±0.000 |
| MuFL | 3 | 6.6±0.4 | 1955±104 | 1.39±0.030 | 0.07±0.005 | 0.22±0.008 | 0.12±0.002 | 0.08±0.002 | 0.05±0.003 | 0.14±0.001 | 0.32±0.011 | 0.38±0.001 | 0.02±0.000 |
| TAG | 4 | 15.8±2.4 | 4639±717 | 1.44±0.007 | 0.07±0.003 | 0.24±0.002 | 0.11±0.001 | 0.10±0.002 | 0.03±0.004 | 0.14±0.000 | 0.35±0.003 | 0.39±0.001 | 0.02±0.000 |
| MuFL | 4 | 7.5±0.3 | 2201±94 | 1.40±0.027 | 0.06±0.004 | 0.22±0.008 | 0.12±0.003 | 0.08±0.002 | 0.05±0.001 | 0.14±0.001 | 0.32±0.011 | 0.39±0.001 | 0.02±0.001 |
| MuFL | 5 | 8.3±0.4 | 2439±105 | 1.40±0.028 | 0.06±0.004 | 0.22±0.008 | 0.12±0.003 | 0.08±0.002 | 0.05±0.000 | 0.14±0.002 | 0.32±0.011 | 0.39±0.001 | 0.02±0.001 |

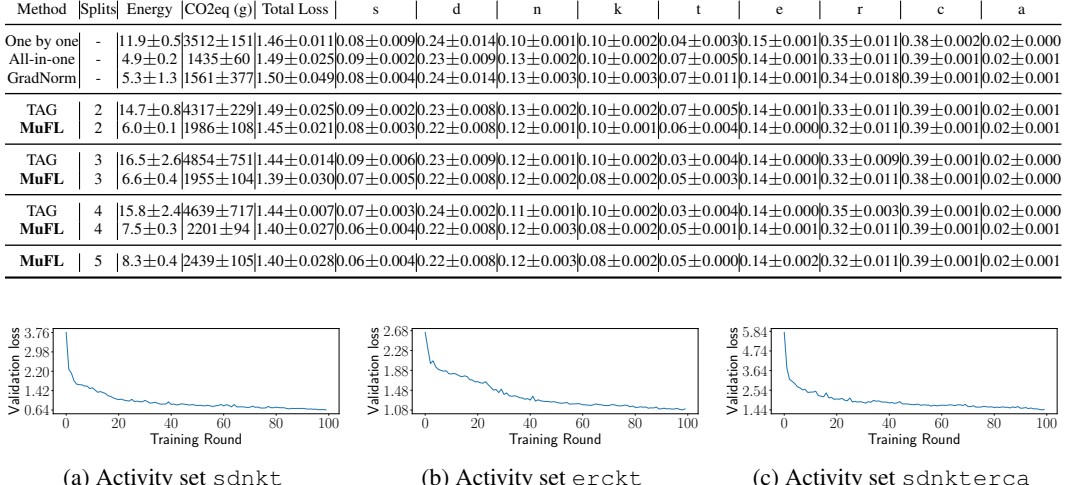

(a) Activity set `sdnkt`      (b) Activity set `erckt`      (c) Activity set `sdnkterca`

Figure 12: Changes of validation loss over the course of training on activity sets: (a) `sdnkt`, (b) `erckt`, and (c) `sdnkterca`. Validation loss converges as training proceeds.

Table 6: Activity splitting results of TAG (Fifty et al., 2021) and MuFL on activity sets `sdnkt`, `erckt`, and `sdnkterca`. Activities of each split is separated by a comma.

| Method | Activity Set | Two Splits | Three Splits | Four Splits | Five Splits |
|---|---|---|---|---|---|
| TAG | sdnkt | sdn,kt | sd,dn,kt | sd,sdn,dn,kt | - |
| MuFL | sdnkt | sdn,kt | sdn,k,t | sd,n,k,t | s,d,n,k,t |
| TAG | erckt | er,rckt | er,kt,rc | er,kt,rc,rt | - |
| MuFL | erckt | er,ckt | er,c,kt | er,c,k,t | e,r,c,k,t |
| TAG | sdnkterca | sdnkterca,dr | sdnerc,dr,kta | sc,dr,ne,kta | - |
| MuFL | sdnkterca | snkteac,dr | snec,dr,kta | sn,dr,ka,etc | sn,dr,ka,e,tc |

**Changes of Vadiation Loss** Figure 12 presents validation losses over the course of all-in-one training of three training activity sets `sdnkt`, `erckt`, and `sdnkterca`. It shows that validation losses converge as training proceeds.

**Splitting Results of Various Methods** We provide results of activity splitting of TAG (Fifty et al., 2021) and MuFL in Table 6. For hierarchical splitting, they further split into three splits from the results of two splits. In particular, the results of hierarchical splitting of `erckt` and `sdnkterca` are the same as their three splits. The hierarchical splitting result of `sdnkt` is from {`sdn,kt`} to {`sd,n,kt`} as the hierarchical splitting further divides the split with more training activities (`sdn`).

Besides, Table 7 presents the splitting results of the optimal and worst splits. They are not identical due to variances in multiple runs of experiments. We report the mean and standard deviation of test losses of the optimal splits and the worst splits in Table 1. The large variances of the optimal and worst splits suggest the instability of splitting by measuring the performances of training from scratch in the FL settings and show the advantage of our methods in obtaining stable splits.

**Dataset Size and Performance** The dataset size of activity set `sdnkt` is around 315GB in our experiments, compared to 2.4TB of dataset used in experiments of TAG (Fifty et al., 2021). The test loss of ours (0.512 in Table 8), however, is better than the optimal one in TAG (Fifty et al., 2021) (0.5246). This back-of-the-envelope comparison indicates the potential to extend our approaches to multi-task learning. Besides, it could also suggest that our data size is sufficient for evaluation.

Table 7: Results of the optimal and worst splits in three runs of experiments. They are not identical due to variances in three runs of experiments.

| Activity Set | Splits | Optimal Splits | | | Worst Splits | | |
|---|---|---|---|---|---|---|---|
| `sdnkt` | 2 | dk,snt | sn,dkt | nt,sdk | st,dnk | st,dnk | st,dnk |
| | 3 | t,sn,dk | k,t,sdn | d,sn,kt | d,st,nk | d,st,nk | s,dt,nk |
| `erckt` | 2 | r,eckt | t,erck | et,rck | rk,ect | ek,rct | e,rckt |
| | 3 | r,ec,kt | r,t,eck | r,ec,kt | c,e,rk | e,k,rct | e,rt,ck |

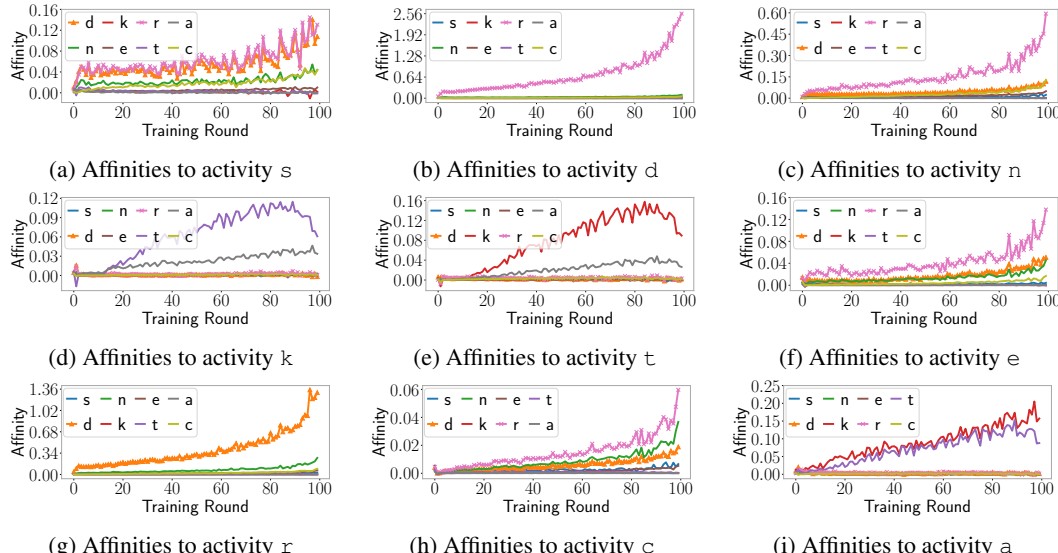

(a) Affinities to activity `s`  (b) Affinities to activity `d`  (c) Affinities to activity `n`

(d) Affinities to activity `k`  (e) Affinities to activity `t`  (f) Affinities to activity `e`

(g) Affinities to activity `r`  (h) Affinities to activity `c`  (i) Affinities to activity `a`

Figure 13: Changes of affinity scores of one activity to the other over the course of training on activity set `sdnkterca`. The trends of affinities emerge at the early stage of training.

**Impact of Affinity Computation Frequency** $f$   The frequency of computing affinities in Equation 2 determines the amount of extra needed computation. We use $f = 5$ and compute affinities for the first ten rounds for all experiments because the trend of affinities emerges in the early stage of training in Figure 13. It would increase the computation of all-in-one training by around 2%, which is already factored into the energy consumption computation in previous experiments. The results in Table 3, 4, and 5 show that MuFL is effective with this setting and the amount of computation is acceptable.

**Impact of Local Epoch**   Figure 14a show the impact of local epoch $E$ on activity sets `sdnkt`, `erckt`, and `sdnkterca`. They complement results of activity set `sdnkt` in Figure 5a. Larger $E$ could lead to better performance with fixed $R = 100$. It is especially effective when increasing $E = 1$ to $E = 2$, but further increasing $E$ could degrade the performance. It indicates that simply increasing computation has limited capability to improve performance.

**Impact of The Number of Selected Clients**   Figure 14b compares the performance of different numbers of selected clients $K = \{2, 4, 6, 8, 16\}$ on three activity sets `sdnkt`, `erckt`, and `sdnkterca`. It complements results in Figure 5b. The results on three activity sets are similar; increasing $K$ reduces losses, but the marginal benefit decreases as $K$ increases.

The majority of experiments in this study are conducted with $K = 4$. We next analyze the impact of $K$ in MuFL with results of two splits on activity set `sdnkt` in Table 8. The results indicate that MuFL is still effective with $K = 8$, which outperforms $K = 4$ and all-in-one training.

**Standalone Training**   Standalone training refers to training using data of each client independently. Figure 15a shows the test loss distribution of thirty-two clients used in experiments. The client ID corresponds to the dataset size distribution in Figure 7. These results suggest that larger data sizes of

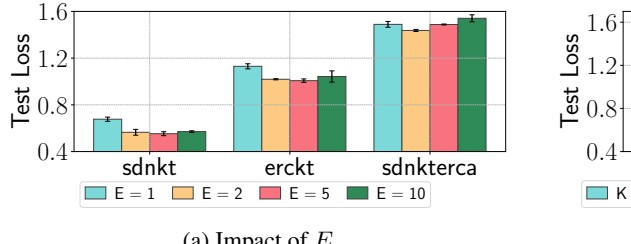

(a) Impact of $E$

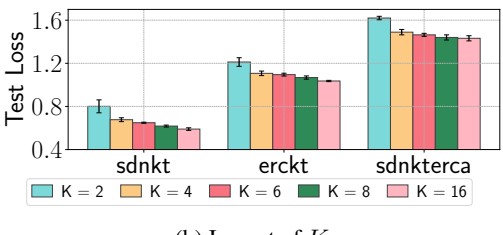

(b) Impact of $K$

Figure 14: Analysis of the impact of local epoch $E$ and impact of the number of selected clients $K$. Larger $E$ (with fixed $R = 100$) and $K$ requires higher computation. They could reduce losses, but the marginal benefit decreases as computation increases.

Table 8: Comparison of test loss using different numbers of selected clients $K$. MuFL achieves even better performance on $K = 8$.

|  | K | Total Loss | s | d | n | k | t |
|---|---|---|---|---|---|---|---|
| All-in-one | 4 | 0.677 | 0.087 | 0.246 | 0.136 | 0.126 | 0.083 |
| All-in-one | 8 | 0.618 | 0.076 | 0.227 | 0.130 | 0.109 | 0.077 |
| MuFL (two splits) | 4 | 0.578 | 0.069 | 0.231 | 0.124 | 0.102 | 0.052 |
| MuFL (two splits) | 8 | **0.512** | **0.060** | **0.202** | **0.117** | **0.083** | **0.048** |

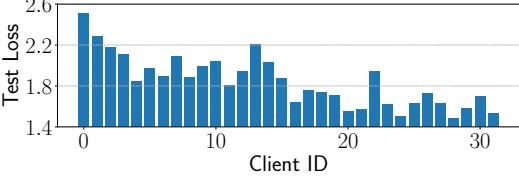

(a) Test loss distribution of standalone training

| Methods | Test Loss |
|---|---|
| Standalone | $1.842 \pm 0.248$ |
| All-in-one | $0.677 \pm 0.018$ |
| MuFL | $0.548 \pm 0.001$ |

(b) Test loss comparison

Figure 15: Performance (test loss) of standalone training that conducts training using data in each client independently: (a) shows the test loss distribution of these thirty-two clients. (b) compares test losses of standalone training and FL methods. We run the experiments on activity set `sdnkt`.

clients may not lead to higher performance. Figure 15b compares test losses of standalone training and federated learning methods. Either all-in-one or our MuFL greatly outperforms standalone training. It suggests the significance of federated learning when data are not sharable among clients.

# D  ALGORITHM

---

**Algorithm 1** Our Proposed Smart Multi-tenant FL System (MuFL)

---

1: **Input:** training activities $\mathcal{A} = \{\alpha_1, \alpha_2, \dots, \alpha_n\}$, a set of available clients $\mathcal{C}$, number of selected clients $K$, local epoch $E$, aggregation weight of client $k$ $p_k$, total training rounds $R$, all-in-one training rounds $R_0$, the number of splits $x$, frequency of computing affinities $f$, batch size $B$

2: **Output:** models $\mathcal{W} = \{\omega_1, \omega_2, \dots, \omega_n\}$

3:

4: **ServerExecution:**

5: Consolidate $\mathcal{A}$ into $\alpha_0$ with a multi-task model $\nu^0 = \{\theta_s\} \cup \{\theta_{\alpha_i} | \alpha_i \in \mathcal{A}\}$ ▷ *Act. consolidation*

6: Initialize $\nu^0$, i.e. initialize $\omega_i = \{\theta_s\} \cup \{\theta_{\alpha_i}\}$ for $i \in \mathcal{N} = \{1, 2, \dots, n\}$

7: **for** *each round* $r = 0, 1, ..., R_0 - 1$ **do**

8:     $\mathcal{C}^r \leftarrow$ (Randomly select K clients from $\mathcal{C}$)

9:     **for** *client* $k \in \mathcal{C}^r$ *in parallel* **do**

10:         $\nu^{r,k}, \hat{\mathcal{S}}^{r,k}_{\alpha_i \to \alpha_j} \leftarrow \underline{\textbf{Client}}(\nu^r, \mathcal{A}, f)$

11:     **end for**

12:     $\nu^{r+1} \leftarrow \sum\limits_{k \in \mathcal{C}^r} p_k \nu^{r,k}$

13:     $\hat{\mathcal{S}}^r_{\alpha_i \to \alpha_j} \leftarrow \frac{1}{K} \sum\limits_{k \in \mathcal{C}^r} \hat{\mathcal{S}}^{r,k}_{\alpha_i \to \alpha_j}$

14: **end for**

15: Compute the values of $\mathcal{S}^r_{\alpha_i \to \alpha_i}, \forall \alpha_i \in \mathcal{A}$, using Equation 3     ▷ *Compute affinity scores*

16: Compute a disjoint partition set $I$ of activities $\mathcal{A}$ for $x$ splits $\{\mathcal{A}_j | j \in I\}$ that maximizes $\mathcal{S}^r_{\alpha_i}$ using affinity scores $\mathcal{S}^r_{\alpha_i \to \alpha_j}, \forall \alpha_i \in \mathcal{A}$ and $\forall \alpha_j \in \mathcal{A}$     ▷ *Activity splitting*

17: **for** *each element* $j \in I$ **do**     ▷ *Schedule to train sequentially*

18:     Initialize $\nu_j = \{\theta^j_s\} \cup \{\theta_{\alpha_i} | \alpha_i \in \mathcal{A}_j\}$ with parameters of $\nu^{R_0}$

19:     **for** *each round* $r = 0, 1, ..., R - R_0 - 1$ **do**

20:         $\mathcal{C}^r \leftarrow$ (Random select K from $\mathcal{C}$)

21:         **for** *client* $k \in \mathcal{C}^r$ *in parallel* **do**

22:             $\nu^{r,k}_j, \_ \leftarrow \text{Client}(\nu^r_j, \mathcal{A}_j, 0)$

23:         **end for**

24:         $\nu^{r+1}_j \leftarrow \sum\limits_{k \in \mathcal{C}^r} p_k \nu^{r,k}_j$

25:     **end for**

26: **end for**

27: Reconstruct $\mathcal{W} = \{\omega_1, \omega_2, \dots, \omega_n\}$ from $\{\nu_j | j \in I\}$ by matching training activities

28: **Return** $\mathcal{W}$

29:

30: **Client** $(\nu, \mathcal{A}, f)$:

31: $T = \lfloor \frac{B}{f} \rfloor$ **if** $f \neq 0$ **else** $0$

32: **for** *local epoch* $e = 1, ..., E$ **do**

33:     Update model parameters $\nu$ with respect to training activities $\mathcal{A}$

34:     **for** each time-step $t = 1, 2, ..., T$ (every $f$ batches) **do**

35:         $\forall \alpha_i \in \mathcal{A}$ and $\forall \alpha_j \in \mathcal{A}$, compute affinities of $\mathcal{S}^t_{\alpha_i \to \alpha_j}$ using Equation 2

36:     **end for**

37: **end for**

38: $\hat{\mathcal{S}}_{\alpha_i \to \alpha_j} = \frac{1}{ET} \sum\limits_{e=1}^{E} \sum\limits_{t=1}^{T} \mathcal{S}^t_{\alpha_i \to \alpha_j}, \forall \alpha_i \in \mathcal{A}$ and $\forall \alpha_j \in \mathcal{A}$

39: **Return** $\nu, \hat{\mathcal{S}}_{\alpha_i \to \alpha_j}$

---

