# OpenReview forum: "Smart Multi-tenant Federated Learning"
_ICLR.cc/2023/Conference — Submitted to ICLR 2023_

### Official Review · Reviewer_icP7 · 2022-10-24

**Confidence:** 3
**Correctness:** 3
**Technical Novelty And Significance:** 2
**Empirical Novelty And Significance:** 2
**Recommendation:** 3

**Clarity, Quality, Novelty And Reproducibility:**

*Clarity: the paper is overall clear
*Novelty: the novelty seems limited
*Reproducibility: no code is provided, so this is unknown

**Strength And Weaknesses:**

**Strength**:
- The general idea proposed by this work is relatively easy to follow.
- The problem of multi-tenant federated learning is interesting. The paper provides an easy-to-understand summary of different types of multi-tenant federated learning.

**Weakness and questions**:
- This contribution of this work seems somewhat limited. The challenge of the problem is not too clear, the performance gain seems be coming from multi-task training which is from existing literature. It seems to be a directly application.
- For the proposed method, the hyper parameters (e.g. R0) seems to be hard to select, and it is more challenging as you have more different types of tasks which might be unknown in advanced. To support multi-tenant, it would be more preferrable to have a method that can automatically selects these hyper parameters.
- It's unclear how the affinity score is obtained. If the training are done in groups (splits) or all-in-one, then don't they share the same model, how do you measure the affinity score between them? Do you need to do additional computation to get that pairwise affinity score. This splitting decision seems to be exponential related to the number of tasks because you need do pairwise comparison and group selection. It's unclear how this is done.


**Summary Of The Paper:**

This work is looking into the multi-tenant federated learning problem, which is considering training multiple federated learning tasks at the same time. The goal of this work is to minimize power consumption while maximizing the performance of training results. This work proposes using a multiple-task share-model technique followed by a splitting method to minimize training loss while reducing power consumption.

**Summary Of The Review:**

Please see above. Overall, I think this paper needs more work to improve its novelty and contribution.

---

> ### Author Response · Authors · 2022-11-13
> **Response to Reviewer icP7**
>
>
> Thank you for reviewing the paper and providing feedback! We have responded below to individual questions from your review.
>
> **Q1**: This contribution of this work seems somewhat limited. The challenge of the problem is not too clear, the performance gain seems be coming from multi-task training which is from existing literature. It seems to be a directly application.
>
> **A1**: Thank you for raising the concern. However, we would like to emphasize that our work is not a direct application of existing literature. Firstly, directly applying multi-task training methods (All-in-one, GradNorm, HOA, and TAG) does not achieve satisfactory results, as shown in Figure 2. Secondly, the reviewer icP7 might have overlooked the last paragraph of Section 4.2, where we explain the differences between our method and the existing method (TAG) in detail. Some reviewers pointed out that the problem of the work is novel (Reviewer CNLB and hdrV) and clearly motivated (Reviewer hdrV). We would like to further clarify that it is challenging to run multiple FL training activities simultaneously on edge devices, because the edge devices have tight resource constraints while deep learning is resource-intensive (2nd paragraph of introduction). Our work is also the first work that attempts to apply multi-task training methods to solve this problem.
>
> **Q2**: For the proposed method, the hyper parameters (e.g. R0) seems to be hard to select, and it is more challenging as you have more different types of tasks which might be unknown in advanced. To support multi-tenant, it would be more preferable to have a method that can automatically selects these hyper parameters.
>
> **A2**: To the best of our knowledge, we are the first work investigating multi-tenant FL in-depth and have proved that our method works well with extensive empirical studies. Our extensive empirical study suggests that R0 = {20, 30, 40} are generally good choices. Proposing a new method to automatically select hyperparameters for different activity sets can be a total new work. For example, the pioneering work in FL, Federated Averaging (FedAvg) [1], proposes to train multiple local epochs E instead of one epoch in local training. The best E is different on different settings. A follow up work proposes to dynamically control the hyperparameter to best fit resource-constrainted scenarios [2].
>
> [1] McMahan, Brendan, et al. "Communication-efficient learning of deep networks from decentralized data." AISTATS 2017.
>
> [2] Wang, Shiqiang, et al. "When edge meets learning: Adaptive control for resource-constrained distributed machine learning." *IEEE INFOCOM 2018*.
>
> **Q3**: It's unclear how the affinity score is obtained. If the training are done in groups (splits) or all-in-one, then don't they share the same model, how do you measure the affinity score between them? Do you need to do additional computation to get that pairwise affinity score. This splitting decision seems to be exponential related to the number of tasks because you need do pairwise comparison and group selection. It's unclear how this is done.
>
> **A3**: The affinity score is computed using Equation (2) during the first 10 rounds of all-in-one training. It consumes around an additional 1% of all-in-one training computation, regardless of the number of tasks. We would like to also clarify that the splitting decision is not *exponential* to the number of tasks. It is similar to the problem of *Stirling numbers of the second kind*. The computation complexity of it is *O(nk)* on dynamic programming implementation, where n is the number of tasks and k is the number of splits. It only takes seconds to compute even for a set of 9 training activities. We would also want to highlight that the energy consumption of our method in Figure 2 has factored in all the computations.
>
> **Q4**: Reproducibility: no code is provided, so this is unknown.
>
> **A4**: Thanks for raising the concern. We are still in the internal processes and will release the codes in the future.

---

> > ### Comment · Reviewer_icP7 · 2022-12-12
> > **Reply to authors**
> >
> > I would like to thank the authors for their response. However, I am still leaning toward keeping my current score. The proposed problem is interesting, but I agree with other reviewers that the contribution and novelty of this work are still somewhat limit.

---

### Official Review · Reviewer_hdrV · 2022-10-25

**Confidence:** 3
**Correctness:** 4
**Technical Novelty And Significance:** 3
**Empirical Novelty And Significance:** 3
**Recommendation:** 5

**Clarity, Quality, Novelty And Reproducibility:**

This paper has clear motivation of the novel setting. The proposed method is clearly introduced. This paper has comprehensive experiments.

**Strength And Weaknesses:**

Strength:

1 This paper proposed a novel setting which is multi-tenant federated learning. It aims to solve the multi-tenant problem in federated learning setting. Four different scenarios are introduced.

2 One vanilla solution and the specifically proposed pipeline are described.

3 Extensive experiments demonstrate the effectiveness and efficiency of the proposed model compared with baselines.

Weaknesses:

1 Due to the novel setting of the problem, the baselines are also the vanilla approach. The significance of the performance improvement.

2 It is not common (general requirements) in edge/device sides to obtain all related and multiple data formats in the same time and to conduct the multi-tenant training. More potential applications/samples are recommended to discuss in the introduction section.


**Summary Of The Paper:**

This paper proposed a novel setting, which is the multi-tenant federated learning especially in edge computing scenario. Specifically, multi-tenant learning aims to optimize/train models which are used for multiple tasks or optimization goals. Federated learning aims to explore subset of the training data for computational resources and private considerations. There is no well explorations which consider both of these requirements, while it is a practical problem in general applications especially in recent years. To this end, this paper proposed 1) a novel setting of multi-tenant federated learning, 2) a vanilla solution of this task, and 3) specifically designed solutions for this task. Extensive experiments are conducted to show the effectiveness and efficiency of the proposed model compared with current general solutions.

**Summary Of The Review:**

This paper proposed a novel setting of multi-tenant federated learning and provides a solution for this problem.

---

> ### Author Response · Authors · 2022-11-13
> **Response to Reviewer hdrV**
>
>
> Thank you for reviewing the paper; we appreciate your positive feedback and suggestions for improvement! We have responded to your concerns below.
>
> **Q1**: Due to the novel setting of the problem, the baselines are also the vanilla approach. The significance of the performance improvement.
>
> **A1**: Thank you for raising this point. Since we are the first work investigating multi-tenant FL in-depth, we define the vanilla approach (one-by-one). Besides the vanilla approach, our approach is also superior to existing multi-task learning methods (TAG and HOA) and simply applying existing FL optimization (FedProx).
>
> **Q2**: It is not common (general requirements) in edge/device sides to obtain all related and multiple data formats in the same time and to conduct the multi-tenant training. More potential applications/samples are recommended to discuss in the introduction section.
>
> **A2**: Thank you for the constructive comment. We have followed your advice to add discussion of potential applications in the introduction and Appendix A.

---

> > ### Comment · Reviewer_hdrV · 2022-11-27
> > **Followup**
> >
> > I have carefully read the reviews from other reviewers and the response from author. Thanks for the author's response of the questions and comments.
> >
> > I lean to host negative opinion of this submission. And the main reasons are:
> >
> > $\cdot$ I agree with other reviews' comments that the level of novelty is not convincing  even through it is claimed the first work in this setting. However, the setting is not that convincing in real-world applications.
> >
> > $\cdot$ As mentioned above, I also raised the question about the practical scenario of the setting, while I'm not convinced by the response since it is not clear about how these applications exactly use the data from multiple modalities.

---

### Official Review · Reviewer_NZnW · 2022-11-03

**Confidence:** 4
**Correctness:** 3
**Technical Novelty And Significance:** 2
**Empirical Novelty And Significance:** 2
**Recommendation:** 3

**Clarity, Quality, Novelty And Reproducibility:**

- Clarity: the paper is overall clear. Minor: $\mathcal{C}$ not defined before first used in Sec 3.2
- Novelty: the novelty seems limited
- Reproducibility: All hyperparameters and configuration is provided in appendix B

**Strength And Weaknesses:**

**Strength**
- Simplicity of proposed approach
- Easy to follow the paper

**Weakness**

- Limited empirical evaluation and significance of the improvements is not clear (both in terms of test loss and energy)
- The source of minor improvement in test loss is also not clear: Maybe the improvements are just originating from multi-task learning due to sharing between multiple activities

- Limited applicability of the setting: It is assumed same server or same service provider is responsible for all the activities in multi-tenant FL. What if server for different activities are different? A much more practical and common scenario would be when the service provider of the keyboard app and camera app are different, i.e. keyboard server can be different from camera server. How would the proposed MuFL be beneficial in that case?

**Summary Of The Paper:**

This submission is considering if multiple federated learning tasks on a single device can be coordinated and consolidated to ease resource constrain and energy consumption. A multi-tenant FL system called MuFL is proposed that 1) Initially merges "similar" training activities into one activity with a multi-task architecture; 2) After some rounds of training, splits back the activities it into groups to have a better synergy. On two dataset empirical studies are carried out which show marginal benefits.

**Summary Of The Review:**

Even though important problem setup, the submission has limited technical innovation and marginal empirical improvements, thereby unfortunately warranting a low overall review score.

---

> ### Author Response · Authors · 2022-11-13
> **Response to Reviewer NZnW**
>
>
> Thank you for reviewing the paper and providing feedback! We have responded below to individual questions from your review.
>
> **Q1**: Limited empirical evaluation and significance of the improvements is not clear (both in terms of test loss and energy)
>
> **A1**: We have conducted extensive empirical evaluations of our method on three activity sets (~27,765 GPU hours). It would be helpful if the reviewer can specify the expected evaluation. As for the concern on the significance of the improvements, our method reduces test loss by more than 0.05 compared with One-by-one and more than 0.1 compared with All-in-one, which is on par with (even better than) other works [1][2] on the same evaluation metric. Also, our method consumes ~40% less energy.
>
> [1] Standley, Trevor, et al. "Which tasks should be learned together in multi-task learning?." ICML 2020.
>
> [2] Fifty, Chris, et al. "Efficiently identifying task groupings for multi-task learning." NeurIPS 2021.
>
> **Q2**: The source of minor improvement in test loss is also not clear: Maybe the improvements are just originating from multi-task learning due to sharing between multiple activities
>
> **A2**: Thank you for raising the concern. As discussed in Q1, the improvement in test loss is not minor. The source of improvement comes from two parts: 1) the reasonable grouping of training activities such that they can resonate well in training and 2) the method of training from initialization.
>
> **Q3**: Limited applicability of the setting: It is assumed same server or same service provider is responsible for all the activities in multi-tenant FL. What if server for different activities are different? A much more practical and common scenario would be when the service provider of the keyboard app and camera app are different, i.e. keyboard server can be different from camera server. How would the proposed MuFL be beneficial in that case?
>
> **A3**: We would like to clarify that the scenario, where different servers are for different activities, is out of the scope of this paper. The common assumption of multi-tenancy of FL is that training activities are based on the same server [3]. Considering different servers for different activities would be an interesting scenario for future work, but it is out of the scope of this work. Besides, *we would like to emphasize that our method could potentially benefit many real-world applications*. For example, autonomous vehicles relate to multiple computer vision (CV) tasks, including object detection, tracking, and semantic segmentation; smart city surveillance cameras associate with various CV tasks, such as crowd counting, object detection, and person re-identification; voice assistant applications like Apple Siri and Google Assistant need multiple automatic speech recognition (ASR) tasks, including word confidence, word deletion, and utterance confidence; smart-manufacturing robots need several CV tasks such as object detection, object grasp point detection, and object pose estimation. There are many more examples. We have also revised the manuscript to emphasize the potential applications of our method.[3] Bonawitz, Keith, et al. (Google). "Towards federated learning at scale: System design." MLSys 2019.
>
> **Q4**: Minor: C not defined before first used in Sec 3.2
>
> **A4**: Thank you for raising the concern. We have defined $\mathcal{C}$ in Sec 3.1.

---

### Official Review · Reviewer_CNLB · 2022-11-03

**Confidence:** 4
**Correctness:** 3
**Technical Novelty And Significance:** 3
**Empirical Novelty And Significance:** 3
**Recommendation:** 3

**Clarity, Quality, Novelty And Reproducibility:**

**Clarity**

The paper is well-written and reader-friendly. The tables do look a bit off, but that could be easily fixed, and the paper is readable.

**Reproducibility**

I believe the work lacks significantly in reproducibility. Although the basic hyperparameters such as number of splits, local epochs have been provided. The paper does not go into the network structure used, the loss functions, the learning rate and other crucial details that are required to replicate the work.

**Novelty**
The authors try to offer an approach to train multiple activities on a resource-contrained device.


**Strength And Weaknesses:**

**Strengths**
* The paper proposes a new domain/problem which could be interesting to AI researchers.
* The paper has ample experiments that there are no issues with pareto-optimality, etc. as usually faced in meta-learning and multi-task learning problems.

**Weaknesses**
* I believe the implementation could be discussed in more detail. The reader is left wondering what the loss was. How come there has been no non-pareto-optimal solutions when training with different splits?
* Although the problem is interesting, I am afraid the limitation of the approach is very closely tied to privacy. For example, suppose a device has training activities related to two different companies/entities. It might not be possible to force the two networks to be same while training, or even train with the same data due to data leakage, privacy concerns, etc. This highly weakens the potential impact of the work. If I have misunderstood this part, I hope the authors could correct me here.
* Although the authors talk about 4 different scenarios in Section 3.2, I believe all experiments could be classified under scenario 1, and the approach has not been shown to be applicable to all 4 scenarios.
* In section 5.5, the authors show that increasing the number of epochs seems to harm the overall test loss. I would assume this is a classical scenario of overfitting, but I hope the authors could explain this. Especially since the exact formulation of the loss used for training is not detailed, this is hard to guess.
* This is more of a question: In Table 3, why is the loss of one-by-one higher than the loss of all-in-one for the task 'd'? This intuitively should not be the case since the all-in-one will simplicity come with more noise.

**Summary Of The Paper:**

The paper presents a method to train multiple simultaneous activities on decentralized edge devices with budget constraints. Most prevailing approaches aim to do this by training tasks one at a time. However, the paper proposes a solution of splitting the tasks into groups and learning jointly. This enables sharing of information as well as an improved efficiency in terms of energy consumption.



**Summary Of The Review:**

Although the results are interesting, it is difficult to truly understand the paper without the complete training regime. The lack of the loss function used is a deterrent, in my view. Besides that, I feel the approach is limited as it can only be used in Scenario 1 (as per Section 3.2).

---

> ### Author Response · Authors · 2022-11-13
> **Response to Reviewer CNLB  (1/2)**
>
>
> Thank you for reviewing the paper and providing feedback! We have responded below to individual questions from your review.
>
> **Q1**: I believe the implementation could be discussed in more detail. The reader is left wondering what the loss was.
>
> **A1**: Thank you for the suggestion. We have followed your advice to describe the loss of All-in-one training in the manuscript and the losses used in experiments in Appendix B. We would like to highlight that our method is not tied to specific losses of training activities -- different training activities can use different loss functions.
>
> **Q2**: How come there has been no non-pareto-optimal solutions when training with different splits?
>
> **A2**: Generally, there is a trade-off between test loss and energy consumption with different splits. More splits need more computation, i.e. consumes more energy, but they could lead to better test loss as shown in Figure 2.
>
> **Q3**: The limitation of the approach is very closely tied to privacy, e.g. a device has training activities related to two different companies/entities.
>
> **A3**: Thank you for raising the concern. **Firstly,** it is more common that the cross-device FL is applied within a company; plenty of application scenarios within a company could be optimized with our approach. Our approach can be directly applied to real-world applications like smart-manufacturing robots, which needs to perform several CV tasks such as object detection, object grasp point detection, and object pose estimation [1]. Similarly, it can be applied to other robotics systems like household robots (e.g., Amazone Astro) and workplace robots (e.g., Google X Company Everyday Robot). Besides robots, more potential applications include but are not limited to the following examples: voice assistant applications like Apple Siri and Google Assistant need multiple automatic speech recognition (ASR) tasks, including word confidence, word deletion, and utterance confidence; autonomous vehicles relate to multiple computer vision (CV) tasks, including object detection, tracking, and semantic segmentation; smart city surveillance cameras associate with various CV tasks, such as crowd counting, object detection, and person re-identification. We have reflected these applications in the introduction and Appendix A of the manuscript. **Secondly,** extending the multi-tenant FL among multiple entities would impose a collaboration agreement beforehand to prevent data or privacy issues. These companies can adopt our method by simply adding an indicator to the training activities -- indicating the training activities that can be merged and the ones that they want to opt-out merging. As for the users, data is always on their devices, the same as standard FL.
>
> [1] Federated Learning for Robot Picking, https://flairop.com/
>
> **Q4**: 4 different scenarios are introduced in Section 3.2, but all experiments are under scenario 1. The approach is limited as it to be used only in Scenario 1.
>
> **A4**: Thank you for raising the concern. We are the first work discussing multi-tenant FL in-depth, so we would like to raise awareness to the community that there are many challenges in different scenarios of multi-tenantFL; And in this work, we have stated in the manuscript that we primarily focus on Scenario 1. We respectfully disagree that it turns out to be the limitation of the work, especially when optimizing Scenario 1 can already be applied to many potential real-world applications as discussed in answers to Q3. We will adopt the reviewer CNLB’s suggestion and optimize other scenarios in our future work.
>
> **Q5**: Why increasing the number of epochs seems to harm the overall test loss? Is it due to overfitting? It is hard to guess as the loss is not detailed.
>
> **A5**: Thanks for raising the concern. We have provided the loss suggested by the reviewer as described in Q1. We agree with you that the reason could be overfitting the local statistical heterogeneous dataset. The data of each client is indoor images of a building, so training too many epochs could lead to the client overfit to local data and harm the generalizability.
>
> **Q6**: In Table 3, why is the loss of one-by-one higher than the loss of all-in-one for the task 'd'?
>
> **A6:** This is specifically related to task relationships. Intuitively, it is possible that task A is good to train with task B, but task B is not good to train with task A. When training multiple tasks together, the relationship is even more complex. *The reason* why task `d` of all-in-one has lower loss is that task `d ` is beneficial to train with other tasks; while other tasks may not gain from training with task `d`. Due to the complexity of relationships among these training activities, we present activity affinity scores to better select the training activities to group together.

---

> > ### Author Response · Authors · 2022-11-13
> > **Response to Reviewer CNLB (2/2)**
> >
> >
> > **Q7**: Lacks significantly in reproducibility: The paper does not go into the network structure used, the loss functions, the learning rate and other crucial details that are required to replicate the work.
> >
> > **A7**: We would like to invite Reviewer CNLB to revisit Appendix B where we have provided the implementation details, such as network structure and learning rate. Loss functions are also added as suggested, including the codes of the loss functions. We will further release all codes after internal processes.

---

> > > ### Comment · Reviewer_CNLB · 2022-12-06
> > > **Followup**
> > >
> > > Thanks for the author's response of the questions and comments. I have carefully read the response from author. I still do lean towards my current score as my concerns have not been fully addressed as elaborated further below.
> > >
> > > - Thank you for adding the loss. Unfortunately, just saying task A uses cross entropy, task B uses L1 loss gives no information about the training of your model. When you are training on a subsets of tasks (say A and B), what is your loss? A linear summation of the two? This might lead to non-pareto optimal solutions, and this is my major concern.
> > > -  Again, the pareto-optimality solutions I am talking about is across tasks, not between energy and loss.
> > > - I do agree with the other reviewers that the novelty is limited, and it is unclear how practical applications can work when multiple modalities are involved.

---

### Author Response · Authors · 2022-11-13
**General Response**

We thank the reviewers for their thoughtful feedback! We are encouraged they find the problem is novel (Reviewer CNLB and hdrV), interesting (Reviewer hdrV and icP7), and clearly motivated (Reviewer hdrV); Also, the proposed approach has simplicity (Reviewer NZnW) and easy to follow (Reviewer hdrV). Besides, we are glad that reviewers find the approach is supported by ample experiments (Reviewer CNLB and hdrV). We are also encouraged that they agree the paper is clear and easy to follow (all reviewers), as well as well-written and reader-friendly (Reviewer CNLB). We answer reviewers’ specific comments in our responses and have incorporated all feedback in the revised manuscript.

---

### Author Response · Authors · 2022-11-18
**Looking Forward to Your Feedback**

Dear Reviewers,

We sincerely thank you for your time and efforts in reviewing our paper, and appreciate your insightful and constructive comments. Reviewers have two major concerns about the work and we would like to further discuss them here. The two concerns are as followed: 1) the limited applicability and impact as we focus on optimizing one of the four multi-tenant FL scenarios we defined in the work. 2) limited contribution and novelty.

We would like to highlight that this is the first work discussing multi-tenant FL in-depth. Our proposed new method is not simply applying existing methods to the FL setting. It is a new algorithm that we designed specifically to address the challenge of multi-tenant FL that multiple simultaneous training activities could overload edge devices. We also discuss the differences between our method and some potential similar methods in the manuscript in detail.

Since we are the first work in the field of multi-tenant FL, we highlighted that there are many challenges in four different scenarios of multi-tenantFL; And in this work, we have stated in the manuscript that we primarily focus on Scenario 1. We respectfully disagree that it turns out to be the limitation of the work, especially when optimizing Scenario 1 can already be applied to many potential real-world applications. For example, it can be directly applied to real-world applications like smart-manufacturing robots, which need to perform several CV tasks such as object detection, object grasp point detection, and object pose estimation [1]. Similarly, it can be applied to other robotics systems like household robots (e.g., Amazone Astro) and workplace robots (e.g., Google X Company Everyday Robot). Besides robots, more potential applications include but are not limited to the following examples: voice assistant applications like Apple Siri and Google Assistant need multiple automatic speech recognition (ASR) tasks, including word confidence, word deletion, and utterance confidence; autonomous vehicles relate to multiple computer vision (CV) tasks, including object detection, tracking, and semantic segmentation; smart city surveillance cameras associate with various CV tasks, such as crowd counting, object detection, and person re-identification. We have reflected these applications in the introduction and Appendix A of the manuscript.

Lastly, we would like to further emphasize that it is becoming increasingly important and common for edge devices like autonomous vehicles and robots to perform multiple tasks, and we are the first work that shed light on training these multiple tasks in FL together to protect data privacy. We have carefully revised the manuscript by incorporating your suggestions and highlighted them in the revised manuscript. Since the discussion time is ending soon, we would like to kindly remind you to check our response and the revised manuscript. We hope it can address your concerns and look forward to your feedback.

Thanks,

Authors

[1] Federated Learning for Robot Picking, https://flairop.com/

---

### Decision · Program_Chairs · 2023-01-20

**Decision:**

Reject

**Justification For Why Not Higher Score:**

Despite authors responses unfortunately none of the reviewers were convinced about the novelty and technical depth. There are unresolved concerns about how loss  between different tasks interact, and where does the improvement arise from: is it just multi-task learning? Also, the significance of the improvements is not evident. Furthermore, all reviewers are questioning the practical application of the proposed method, but that is of lower importance for an academic conference.

**Justification For Why Not Lower Score:**

N/A

**Metareview: Summary, Strengths And Weaknesses:**

The paper attempts to improve decentralized training on edge devices using federated learning, like reduce energy footprint. In this regards, authors propose to coordinate and consolidate multiple federated learning tasks on a single device. A multi-tenant FL system called MuFL is proposed that 1) Initially merges "similar" training activities into one activity with a multi-task architecture; 2) After some rounds of training, splits back the activities it into groups to have a better synergy. On two dataset empirical studies are carried out which show marginal benefits. We thank the authors and reviewers for actively engaging in discussion and improving the paper like adding loss function details. However, despite authors responses unfortunately none of the reviewers were convinced about the novelty and technical depth. There are unresolved concerns about how loss  between different tasks interact, and where does the improvement arise from: is it just multi-task learning? Also, the significance of the improvements is not evident. Furthermore, all reviewers are questioning the practical application of the proposed method, but that is of lower importance for an academic conference.